# Learning to Handle Complex Constraints
# for Vehicle Routing Problems

**Jieyi Bi**[1], **Yining Ma**[1,†], **Jianan Zhou**[1],
**Wen Song**[2], **Zhiguang Cao**[3], **Yaoxin Wu**[4], **Jie Zhang**[1]

[1]Nanyang Technological University
[2]Shandong University
[3]Singapore Management University
[4]Eindhoven University of Technology

`jieyi001@e.ntu.edu.sg`, `yiningma@u.nus.edu`,
`jianan004@e.ntu.edu.sg`, `wensong@email.sdu.edu.cn`,
`zgcao@smu.edu.sg`, `y.wu2@tue.nl`, `zhangj@ntu.edu.sg`

## Abstract

Vehicle Routing Problems (VRPs) can model many real-world scenarios and often involve complex constraints. While recent neural methods excel in constructing solutions based on feasibility masking, they struggle with handling complex constraints, especially when obtaining the masking itself is NP-hard. In this paper, we propose a novel Proactive Infeasibility Prevention (PIP) framework to advance the capabilities of neural methods towards more complex VRPs. Our PIP integrates the Lagrangian multiplier as a basis to enhance constraint awareness and introduces preventative infeasibility masking to proactively steer the solution construction process. Moreover, we present PIP-D, which employs an auxiliary decoder and two adaptive strategies to learn and predict these tailored masks, potentially enhancing performance while significantly reducing computational costs during training. To verify our PIP designs, we conduct extensive experiments on the highly challenging Traveling Salesman Problem with Time Window (TSPTW), and TSP with Draft Limit (TSPDL) variants under different constraint hardness levels. Notably, our PIP is generic to boost many neural methods, and exhibits both a significant reduction in infeasible rate and a substantial improvement in solution quality.

## 1 Introduction

Vehicle routing problems (VRPs) are NP-hard combinatorial optimization problems with complex constraints that model real-world scenarios, such as logistics [1] and supply chains [2]. For decades, traditional solvers relied on hand-crafted rules for VRP optimization and constraint handling. Recently, the learning-to-optimize community [3] has successfully trained deep neural networks to automatically construct VRP solutions in an end-to-end manner [4–6]. These data-driven neural methods offer greater efficiency and high parallelism for batch optimization, making them favorable alternatives.

In general, neural methods construct VRP solutions by autoregressively sampling a node from its predicted distribution while masking out nodes that would violate constraints to ensure the solution's feasibility. Despite successes (e.g., on TSP and CVRP), this masking mechanism assumes that 1) the feasibility of the entire solution can be properly decomposed into the feasibility of each node selection

---

[†]Yining Ma is the corresponding author.

38th Conference on Neural Information Processing Systems (NeurIPS 2024).

step, and 2) ground truth masks are easily obtainable for each step. However, such assumptions may fail in VRPs (e.g., TSPTW) with complex interdependent constraints among decision variables (i.e., nodes). As will be discussed in Section 4.1, this creates a masking dilemma - considering only the local feasibility of node selections does not guarantee the overall feasibility of the constructed solutions, while computing global feasibility masks that account for future impacts transforms masking itself into another intractable NP-hard problem.

These observations highlight significant gaps in applying recent neural methods to practical VRPs, necessitating research on new constraint-handling frameworks. In the literature, few studies have focused on novel ways of handling feasibility in neural constructive solvers. Although preliminary methods have attempted to mitigate it by relaxing constraints into soft ones [7, 8] or supplementing networks with more feasibility-related features [9], the former is prone to failure when applied to more complex scenarios, while the latter requires problem-specific features and a large supervised learning dataset, limiting its adaptability to broader VRPs. Consequently, neural methods still show limited flexibility, poor feasibility rates and large optimality gaps in solving those complex VRPs.

In this paper, we propose a novel **P**roactive **I**nfeasibility **P**revention (**PIP**) framework to extend the capabilities of neural constructive methods for VRPs with complex interdependent constraints. Our PIP first integrates the Lagrangian multiplier method into the reinforcement learning framework of neural methods, promoting initial constraint awareness and search guidance. To further address the limitations of the Lagrangian multiplier method on complex constraints, we then introduce preventative infeasibility masking to proactively steer the search to (near-)feasible regions during solution construction. By doing so, PIP significantly enhances feasibility rates and reduces optimality gaps. Moreover, to reduce the costs of obtaining preventative infeasibility information during training, we present **PIP-D**, which employs an auxiliary decoder to learn and predict masking information. Our PIP-D also incorporates two adaptive strategies: one to balance infeasible and feasible masking information for different problem hardness, and another to periodically update the model so as to balance training efficiency with prediction accuracy. These advancements enable PIP-D to achieve comparable or even better performance than PIP, particularly on larger and more constrained VRP instances, while significantly reducing computational complexity.

Our contributions are as follows: 1) *Conceptually*, we represent an early work to address and advance the handling of complex interdependent constraints in VRPs, where the original masking loses effectiveness due to the aforementioned dilemma, thereby extending the applications of neural methods to more practical scenarios. 2) *Methodologically*, we propose novel PIP and PIP-D approaches that can boost the capabilities of most constructive neural methods. Specifically, we leverage the Lagrangian multiplier method and introduce preventative infeasibility masking, which is further learned by an auxiliary decoder network with two adaptive strategies, to proactively and efficiently steer the search during solution construction. 3) *Experimentally*, we conduct extensive validation to demonstrate the effectiveness and versatility of PIP across various backbone *models* (i.e., AM [4], POMO [5], and GFACS [10]) and complex VRP *variants* (i.e., TSPTW and TSPDL). Notably, PIP achieves both a significant (up to 93.52%) reduction in infeasible rate and a substantial improvement in solution quality on synthetic and benchmark datasets with different constraint hardness levels.

## 2   Related work

**Neural solvers for VRPs.** Existing literature on learning to optimize VRPs features two primary paradigms: constructive solvers and iterative solvers. *Constructive solvers* learn policies to construct solutions from scratch in an end-to-end manner. Early works introduce Pointer Network to approximate the optimal solution to TSP [11, 12] and CVRP [13] in an autoregressive (AR) way. Among all AR solvers, the attention-based model (AM) [4] represents a milestone in solving a series of VRPs. Later, the policy optimization with multiple optima (POMO) [5] further improves upon AM by considering the symmetry property of VRP solutions. Numerous recent studies have then aimed to further enhance their performance [14–23] and versatility [24–27]. Besides the AR methods, several works construct a heatmap, which indicates the probability distribution of each edge being part of the optimal solution, to solve VRPs in a non-autoregressive (NAR) manner [10, 28–34]. Despite the superior performance on large-scale instances, we note that a recent work [35] questions the effectiveness of heatmap generative methods due to the misalignment of training and testing objectives. Differently, *iterative solvers* learn policies to iteratively refine an initial solution. The policies are often trained in contexts of classic heuristics or meta-heuristics for obtaining more

efficient and effective search components [36–45]. Generally, constructive solvers can efficiently achieve desirable performance levels, whereas iterative solvers hold the potential to search for near-optimal solutions with a prolonged time budget. Additionally, there are also several works studying the scalability [46–52], generalization [53–58], and robustness [59, 60] of neural VRP solvers, and leveraging large language models (LLMs) to optimize VRPs [61–63].

**Constraint handling for VRPs.** Most neural methods for VRPs manage constraints using a feasibility masking mechanism that eliminates actions leading to infeasible solutions during construction or iteration search [4, 28, 32, 41]. However, such a mechanism assumes the availability of accurate masks and often lacks constraint awareness learning during training, which is not always practical or desirable. For example, Zhao et al. [64] highlighted the benefits of learning to modulate agent behaviours in the 3D Bin Packing Problem, and Ma et al. [45] showed that temporary constraint violations could enhance neural iterative solvers. Despite their successes, these approaches are inherently unsuitable for assisting constructive solvers to address the VRPs with complex interdependent constraints studied in this paper. While Tang et al. [8] and Zhang et al. [7] proposed methods to transform hard constraints into soft ones via relaxation techniques and problem redefinition, respectively, they may only be able to yield near-feasible solutions with large infeasible rates for VRPs with complex constraints. More recently, Chen et al. [9] developed a multi-step look-ahead (MUSLA) method specifically tailored for TSPTW, incorporating problem-specific features and a large supervised learning dataset. In contrast, this paper proposes a more flexible and generic PIP framework based on novel ideas of preventative infeasibility masking, learnable decoders, and adaptive strategies to advance a broader range of neural methods without needing labelled training data.

## 3 Preliminaries

In this paper, we mainly consider two VRP variants with complex interdependent constraints (i.e., TSPTW and TSPDL), and neural solvers (i.e., AM [4], POMO [5] and GFACS [10]).

**Problem definitions and notations.** A VRP instance can be defined over a complete graph $\mathcal{G} = \{\mathcal{V}, \mathcal{E}\}$, where $\mathcal{V} = \{v_0, v_1, \ldots, v_n\}$ denotes the node set, and $\mathcal{E} = \{e(v_i, v_j) | v_i, v_j \in \mathcal{V}, i \neq j\}$ denotes the directed edge set among all nodes. The objective is to minimize the total cost (e.g. Euclidean length) of the solution tour. To form a feasible solution, each node in $\mathcal{V}$ should be visited exactly once while respecting problem-specific constraints. We consider two types of VRP constraints that are practical in industry: 1) *Time window constraint:* The arrival time at node $v_i$, denoted as $t_i$, must fall within a customer-specific time window $[l_i, u_i]$. If The vehicle arrives early (i.e., $t_i < l_i$), it must wait until $l_i$; 2) *Draft limit constraint:* Each node $v_i$ represents a port with a non-negative demand $\delta_i$ and a maximum draft $d_i$. We denote the current cumulative load of the freighter at port $v_i$ as $\alpha_i$ in a given solution, which should not exceed the corresponding maximum draft $d_i$ of the port.

**Constructive solvers for VRPs.** Popular neural constructive solvers [4, 5] typically parameterize the policy using an encoder-decoder model with parameter $\theta$, trained with reinforcement learning (RL). Given a VRP instance $\mathcal{G} = \{\mathcal{V}, \mathcal{E}\}$, the features of each node $v_i$ are represented as $f_i^v = \{x_i, y_i, c_i\}$, where $x_i, y_i$ are node coordinates, $c_i$ represents constraint-related features (e.g., $c_i = \{l_i, u_i\}$ for time windows in TSPTW and $c_i = \{\delta_i, d_i\}$ for demand and draft limits in TSPDL). The encoder transforms node features into high-dimensional representation embeddings $h_i$, which, combined with the context of the partial tour, represent the current state. The decoder takes them as inputs and outputs probabilities for candidate nodes (actions). The reward $\mathcal{R}(\tau|\mathcal{G})$ is the negative tour length. The policy $\pi_\theta$ is typically trained using REINFORCE [65] as follows:

$$\nabla \mathcal{L}_{\mathrm{RL}}(\theta|\mathcal{G}) = \frac{1}{K} \sum_{i=1}^{K} (\mathcal{R}(\tau_i|\mathcal{G}) - b(\mathcal{G})) \nabla \log \pi_\theta(\tau_i|\mathcal{G}), \tag{1}$$

where $K$ denotes the number of sampled solutions $\tau_i$ for a given training instance $\mathcal{G}$, and $b(\cdot)$ is a baseline function to reduce the variance. Specifically, the baseline is the reward (negative tour length) of the solution derived greedily in AM or the average reward of sampled solutions $\frac{1}{K} \sum_{i=1}^{K} \mathcal{R}(\tau_i|\mathcal{G})$ in POMO. Notably, POMO stipulates the starting node of each solution for diversification, which, however, may hinder solution feasibility in our studied complex constrained problems. Based on our preliminary experiments, POMO with and without diverse starting nodes achieve around 50.70% and 1.75% infeasible rates on the easy TSPTW-50 datasets, respectively. Therefore, we remove the starting node stipulation in POMO and instead sample $n$ solutions to calculate the baseline.

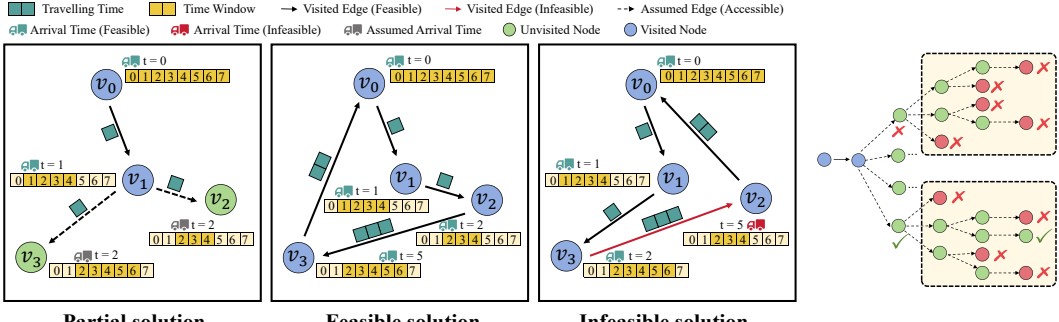

Figure 1: A TSPTW instance to illustrate the malfunction of existing masking mechanism (left three panels) and NP-hardness of obtaining precise infeasible masks (right panel). The orange bar represents the time window $[l_i, u_i]$ for node $v_i$. For the partial solution $v_0 \to v_1$, both $v_2$ and $v_3$ are locally feasible. However, selecting $v_3$ results in the irreversible infeasibility of $v_2$ afterwards.

## 4 Methodology

We now discuss the limitations of existing masking mechanisms in solving VRPs with complex interdependent constraints, followed by a detailed introduction of our PIP and PIP-D frameworks.

### 4.1 Dilemma of feasibility masking

The core of feasibility masking in neural constructive solvers is to filter out invalid actions that violate constraints, based on the assumption that the global feasibility can be decomposed into the feasibility of each node selection step, and that ground truth masks are obtainable for each step. Without loss of generality, we illustrate the dilemma of feasibility masking using a TSPTW example. In TSPTW, nodes are masked out if they have been visited or cannot be visited before their time window closes. However, the feasibility of selecting a node at a particular step impacts the current time, thereby affecting all future selections due to the interdependence of time window constraints. Thus, considering only local feasibility does not guarantee overall feasibility and may lead to irreversible infeasibility. For instance, in a 4-node TSPTW instance with time windows $\{[0,7], [1,4], [2,4], [2,6]\}$ as illustrated in the left panel of Figure 1, there is a feasible solution $\tau = (v_0 \to v_1 \to v_2 \to v_3)$. Yet, with the partial solution $v_0 \to v_1$, both $v_2$ and $v_3$ appear locally feasible. If the solver selects $v_3$, the tour becomes infeasible irreversibly. A potential remedy is to compute global feasibility masks that consider all future possibilities, as illustrated in the right panel of Figure 1. However, this makes masking itself an NP-hard problem, which creates a dilemma between ensuring solution feasibility and managing computational complexity. Note that this dilemma is less critical in CVRPTW, which involves multiple vehicles and routes, providing more flexibility. If one route becomes infeasible, another vehicle departing at time 0 can cover the missed nodes, reducing the impact of constraint interdependencies. However, this issue is severe in TSPTW and other variants like TSPDL.

### 4.2 Guided policy search by PIP

We first formulate the solution construction process of VRP as a Constrained MDP (CMDP) defined by the tuple $(\mathcal{S}, \mathcal{A}, \mathcal{P}, \mathcal{R}, \mathcal{C})$, where $\mathcal{S}$ is the state space, $\mathcal{A}$ is the action space that travels from node $v_i$ to node $v_j$, $\mathcal{R} : \mathcal{S} \times \mathcal{A} \times \mathcal{S}$ is the reward function, $\mathcal{C} : \mathcal{S} \times \mathcal{A} \times \mathcal{S}$ is the constraint violation cost (penalty) function, and $\mathcal{P} : \mathcal{S} \times \mathcal{A} \times \mathcal{S} \to [0,1]$ is the transition probability function. At each time step, the neural solver outputs the probability of all candidate nodes, and selects one to construct a complete solution $\tau$. The objective of CMDP is to learn a policy $\pi_\theta : \mathcal{S} \to \mathcal{P}(\mathcal{A})$ that maximizes the summation of the state-wise reward subject to certain constraints,

$$\max_\theta \mathcal{J}(\pi_\theta) = \mathbb{E}_{\tau \sim \pi_\theta} \left[ \sum_{e(v_i, v_j) \in \tau} \mathcal{R}\left(e\left(v_i, v_j\right)\right) \right],$$

$$\text{s.t. } \pi_\theta \in \Pi_F, \ \Pi_F = \{\pi \in \Pi \,|\, \mathcal{J}_{\mathcal{C}_m}(\pi) \le \kappa_m, \ \forall m \in [1, M]\}, \tag{2}$$

where $\mathcal{J}$ is the expected return of the policy, $\Pi_F$ denotes the set of all feasible policies, $\kappa_m$ represents the boundary of the inequality constraints $\mathcal{C}_m$, and $M$ is the number of constraints. Specifically, a

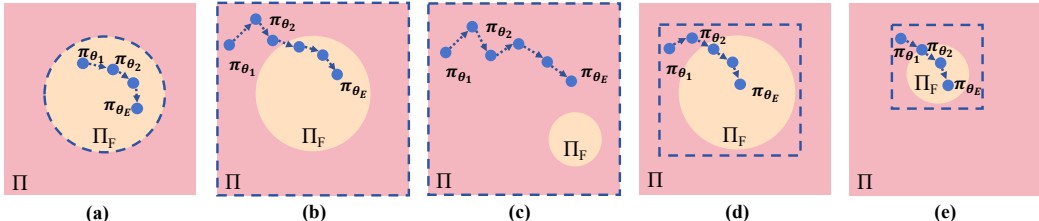

Figure 2: Illustration of policy optimization trajectories on VRP with varying difficulty levels - (a)(b)(d) easy and (c)(e) hard, and different constraint handling schemes - (a) feasibility masking, (b)(c) Lagrangian multiplier, and (d)(e) our PIP. The orange-filled circle denotes the feasible policy space $\Pi_F$, while the dotted frame represents the actual search space of the neural policies $\pi_\theta$.

feasible policy $\pi$ is one whose expected value of constraint violation w.r.t $\mathcal{C}_m$, denoted as $\mathcal{J}_{\mathcal{C}_m}(\pi)$, does not exceed $\kappa_m$. Note that $\kappa_m$ is set to 0 throughout this paper since we consider the hard constraints that do not tolerate any violation. Moreover, we set the reward function $\mathcal{R}$ to the negative value of the Euclidean distance between two nodes, i.e., $\mathcal{R}(e(v_i, v_j)) = -||v_i - v_j||_2$.

By applying feasibility masking, the search is confined to only feasible regions, allowing neural methods to focus solely on the objective function in Eq. (2) without explicitly considering constraint awareness or constraint violations. However, these methods lose effectiveness when such masks are unavailable, leading to inefficient searches in large infeasible regions. To address this, we propose PIP, combining a Lagrangian multiplier for constraint awareness and preventative infeasibility masking to confine the search space to near-feasible regions for complex constrained problems.

**Lagrangian-assisted constraint awareness.** We design a Lagrangian multiplier based method to incorporate constraints $\mathcal{C}$ into the reward function $\mathcal{R}$. Based on the Lagrangian Multiplier Theorem, the CMDP formulation in Eq. (2) is transformed into the following MDP formulation for VRPs:

$$\min_{\lambda \geq 0} \max_{\theta} \mathcal{L}(\lambda, \theta) = \min_{\lambda \geq 0} \max_{\theta} -\mathbb{E}_{\tau \sim \pi_\theta} \left[ \sum_{e(v_i, v_j) \in \tau} ||v_i - v_j||_2 + \sum_{m=1}^{M} \lambda_m \mathcal{J}_{C_m}(\tau) + \mathcal{J}_{\text{IN}} \right], \quad (3)$$

where $\mathcal{L}$ is the Lagrangian function, and $\lambda_m$ is a non-negative Lagrangian multiplier. Generally, the constraint violation term is calculated as the total violation value of all constraints. In TSPTW, $\mathcal{J}_{\text{TW}}(\tau) = \sum_{i=0}^{n} \max(t_i - u_i, 0)$, and in TSPDL, $\mathcal{J}_{\text{DL}}(\tau) = \sum_{i=0}^{n} \max(\alpha_i - d_i, 0)$. Additionally, we introduce the number of infeasible nodes in the solution $\tau$, termed as $\mathcal{J}_{\text{IN}}$, as an extra term in the Lagrangian function for better constraint awareness, which is empirically found to be effective to reduce the infeasibility rate. While Lagrangian relaxation has been explored in neural iterative methods for soft objectives [8], our approach introduces a distinct constraint violation cost function tailored for neural constructive methods and considers fixing the Lagrangian multiplier $\lambda$ (the dual variable) and optimizing the primal variable $\theta$, significantly reducing computational overheads.

**Preventative infeasibility (PI) masking.** As depicted in Figure 2(b), the customized Lagrangian multiplier guides the neural policy towards a potentially feasible and high-quality space using Eq. (3). However, for more complex cases shown in Figure 2(c), neural solvers may still struggle to navigate the large search space. To further improve training efficiency and solution feasibility, we introduce *preventative infeasibility (PI) masking* to proactively avoid selecting infeasible nodes during the solution construction process. As shown in the left panel of Figure 3, if selecting a candidate node (i.e., orange node) results in any remaining candidates (i.e., green node) becoming potentially unvisitable in the next step due to constraint violations, it is marked as infeasible (i.e., red node) since selecting it would cause irreversible future infeasibility (see Appendix A.3 for a detailed example). Note that we employ a simple yet effective one-step PI masking in this paper to balance computational costs without iterating over all future possibilities (which is NP-hard). Together with the customized Lagrangian multiplier, our PIP proactively reduces the search space to a near-feasible domain $\Pi_{\widetilde{F}}$, as shown in Figures 2(d)-(e). Notably, such PIP design is generic and can be applied to enhance most neural constructive solvers for VRPs with complex interdependent constraints.

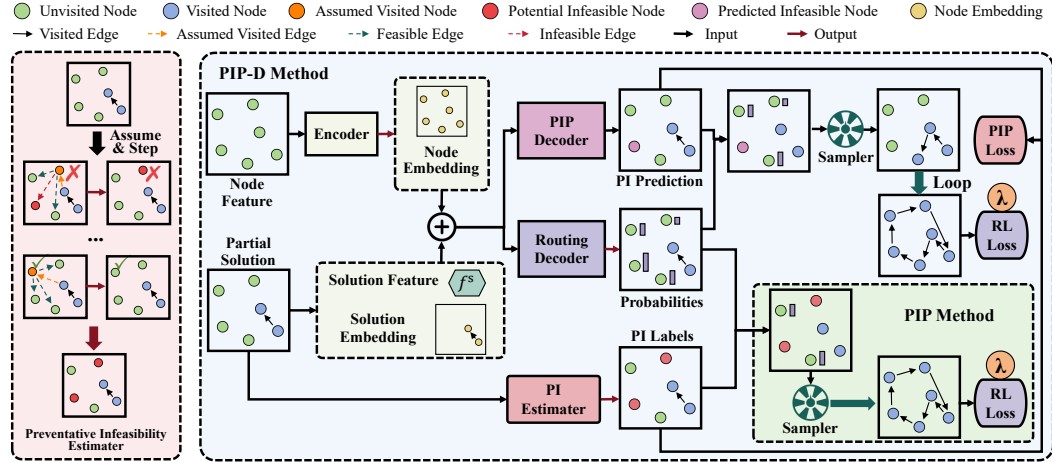

Figure 3: An illustrative overview of our proposed approach: Left - Preventative infeasibility (PI) estimator. Right - PIP (highlighted in green) framework and PIP-D (highlighted in blue) framework.

## 4.3 Learning to prevent infeasibility

Benefiting from the constraint-aware optimization guided by our PIP, neural methods gain enhanced capabilities to address complex constraints, significantly boosting feasibility and optimality. However, acquiring the above PI information introduces extra computational costs (see Section 5). To alleviate this, we propose an auxiliary decoder network to learn and predict these masks, replacing the time-consuming process of generating PI information with a much faster forward pass of the PIP decoder. This further accelerates the training process, resulting in an enhanced version of our PIP framework, termed PIP-D. The overall framework of PIP-D is illustrated in Figure 3.

**Auxiliary PIP decoder.** As presented in Figure 3, we incorporate an auxiliary decoder to learn and predict the PI masks. Our PIP-D simultaneously involves training a routing decoder (the original one) that maximizes the expected reward of solutions in Eq. (3) and a PIP decoder that minimizes the prediction error on the PI masking using a weighted binary cross-entropy loss. The combined loss function is a weighted sum of these two objectives, i.e., $\mathcal{L} = \alpha\mathcal{L}_{\text{RL}} + \beta\mathcal{L}_{\text{PIP}}$. To ensure the generality, the PIP decoder mirrors the architecture of the backbone model and only adjusts the final output layer with Sigmoid activation. More details of our PIP decoder are provided in Appendix B.

**PIP-D training with adaptive strategies.** Nevertheless, efficiently training the PIP decoder together with the routing decoder necessitates effective designs. We address this with two adaptive strategies. Firstly, training the PIP decoder at every gradient step would result in higher computational complexity than the original PIP, countering our goal of reducing training complexity. Hence, we adopt a periodic update strategy that intermittently updates the PIP decoder instead of continuously doing so. This approach is based on the observation that the PI masks recommended by the neural network tend to remain robust over short training periods. Specifically, we first train the PIP decoder with $E_{\text{init}}$ epochs, then periodically update $E_u$ epochs per $E_p$ epochs, and finally conduct $E_l$-epoch updates. In this way, the computational costs are reduced and can be adaptively adjusted. Secondly, we consider balancing feasible and infeasible PI signals for instances with different inherent hardness. Given that the proportion of PI signals identified for feasible and infeasible nodes can vary significantly across different VRP variants with different inherent hardness, we employ a weighted balancing strategy to mitigate the influence of label imbalance [66], which is formulated as follows:

$$\nabla\mathcal{L}_{\text{PIP}}\left(\theta|\mathcal{G}\right) = -\frac{1}{T}\sum_{t=0}^{T}\left(\omega_{\text{infsb}} \cdot g_t \cdot \nabla\log\left(p_\theta\left(g_t\right)\right) + \omega_{\text{fsb}} \cdot \left(1 - g_t\right) \cdot \nabla\log\left(1 - p_\theta\left(g_t\right)\right)\right), \quad (4)$$

where $T$ is the total decoding step to construct a complete solution. The weights of each category are calculated by their corresponding sample number, i.e, $\omega_{\text{infsb}} = \frac{N_{\text{infsb}}+N_{\text{fsb}}}{2N_{\text{infsb}}}, \omega_{\text{fsb}} = \frac{N_{\text{infsb}}+N_{\text{fsb}}}{2N_{\text{fsb}}}$, where $N_{\text{infsb}}$ and $N_{\text{fsb}}$ are the number of infeasible and feasible nodes identified by our PI masking ($g_t$) in a specific decoding step $t$, respectively. Moreover, beyond the above two critical strategies, we explore additional strategies to accelerate the training of the PIP decoder, including fine-tuning and early-stopping techniques, which are discussed in Appendix D.3.

Table 1: Experiments on TSPTW instance with three different hardness†.

| Method | n = 50 Infeasible% Sol.↓ | Inst.↓ | Obj.↓ | Gap↓ | Time↓ | n = 100 Infeasible% Sol.↓ | Inst.↓ | Obj.↓ | Gap↓ | Time↓ |
|---|---|---|---|---|---|---|---|---|---|---|
| **Easy** | | | | | | | | | | |
| LKH3 | 0.00% | 0.00% | 7.31 | 0.00% | 4.6h | 0.00% | 0.00% | 10.21 | 0.00% | 8.5h |
| ORTools | 0.00% | 0.00% | 7.34 | 0.96% | 7h | 0.00% | 0.00% | 10.41 | 1.97% | 14h |
| Greedy-L | 100.00% | 100.00% | / | / | 13.8s | 100.00% | 100.00% | / | / | 1.3m |
| Greedy-C | 0.00% | 0.00% | 26.08 | 257.27% | 4.5s | 0.00% | 0.00% | 52.14 | 411.13% | 12s |
| JAMPR # | / | 0.00% | / | 249.03% | 1.2m | / | 100.00% | / | / | 1.6m |
| OSLA # | / | 11.80% | / | 8.15% | 15.6s | / | / | / | / | / |
| MUSLA # | / | 8.20% | / | 7.32% | 1.3m | / | 18.60% | / | 14.6% | 9.8m |
| MUSLA adapt # | / | 0.10% | / | 5.63% | 7.7m | / | 0.60% | / | 12.01% | 1.1h |
| AM | 100.00% | 100.00% | / | / | 5m | 100.00% | 100.00% | / | / | 21m |
| AM* | 3.46% | 0.22% | 8.02 | 9.82% | 5.2m | 7.87% | 1.49% | 11.84 | 16.07% | 21m |
| AM*+PIP | 0.55% | 0.00% | 7.87 | 7.67% | 10.7m | 0.45% | 0.00% | 11.42 | 11.86% | 1h |
| AM*+PIP-D | 0.51% | 0.00% | 7.91 | 8.19% | 11m | 0.25% | 0.00% | 11.53 | 13.02% | 1h |
| POMO | 100.00% | 100.00% | / | / | 13s | 100.00% | 100.00% | / | / | 21s |
| POMO* | 1.75% | 0.00% | 7.54 | 3.08% | 13s | 2.11% | 0.00% | 10.83 | 6.07% | 21s |
| POMO* + PIP | 0.32% | 0.00% | 7.50 | 2.65% | 15s | 0.15% | 0.00% | 10.57 | 3.53% | 48s |
| POMO* + PIP-D | 0.28% | 0.00% | 7.49 | 2.51% | 15s | 0.06% | 0.00% | 10.66 | 4.39% | 48s |
| **Medium** | | | | | | | | | | |
| LKH3 | 0.00% | 0.00% | 13.02 | 0.00% | 7h | 0.00% | 0.00% | 18.74 | 0.00% | 10.8h |
| ORTools | 15.77% | 15.77% | 13.02 | 0.30% | 5.9h | 0.52% | 0.52% | 19.34 | 3.23% | 13.8h |
| Greedy-L | 100.00% | 100.00% | / | / | 15s | 100.00% | 100.00% | / | / | 1m |
| Greedy-C | 47.52% | 47.52% | 25.33 | 96.43% | 4.2s | 20.34% | 20.34% | 51.62 | 176.07% | 11.4s |
| AM | 100.00% | 100.00% | / | / | 5m | 100.00% | 100.00% | / | / | 21m |
| AM* | 24.84% | 0.27% | 13.81 | 6.11% | 5m | 50.19% | 0.09% | 21.42 | 14.34% | 21m |
| AM*+PIP | 7.62% | 0.35% | 13.68 | 5.06% | 11m | 12.73% | 0.04% | 20.57 | 9.82% | 1h |
| AM*+PIP-D | 11.96% | 0.33% | 13.65 | 4.87% | 11m | 8.80% | 0.02% | 20.80 | 11.03% | 1h |
| POMO | 100.00% | 100.00% | / | / | 13s | 100.00% | 100.00% | / | / | 21s |
| POMO* | 14.92% | 3.77% | 13.68 | 5.23% | 13s | 18.77% | 0.12% | 20.78 | 10.93% | 21s |
| POMO* + PIP | 4.53% | 0.90% | 13.40 | 2.91% | 15s | 3.88% | 0.19% | 19.61 | 4.65% | 48s |
| POMO* + PIP-D | 3.83% | 0.65% | 13.45 | 3.32% | 15s | 3.34% | 0.03% | 19.79 | 5.64% | 48s |
| **Hard** | | | | | | | | | | |
| LKH3 | 0.12% | 0.12% | 25.61 | 0.00% | 7h | 0.07% | 0.07% | 51.24 | 0.00% | 1.4d |
| ORTools | 65.72% | 65.72% | 25.76 | -0.00% | 2.4h | 89.07% | 89.07% | 51.61 | 0.00% | 1.6h |
| Greedy-L | 100.00% | 100.00% | / | / | 21.8s | 100.00% | 100.00% | / | / | 1.3m |
| Greedy-C | 72.55% | 72.55% | 26.39 | 1.53% | 4.5s | 93.38% | 93.38% | 52.95 | 1.43% | 11.1s |
| AM | 100.00% | 100.00% | / | / | 5m | 100.00% | 100.00% | / | / | 21m |
| AM* | 39.87% | 18.88% | 26.08 | 1.425% | 5m | 100.00% | 100.00% | / | / | 21m |
| AM*+PIP | 18.07% | 1.98% | 25.71 | 0.38% | 11m | 41.92% | 16.46% | 51.49 | 0.47% | 1h |
| AM*+PIP-D | 30.39% | 4.40% | 25.80 | 0.67% | 11m | 53.09% | 5.33% | 51.55 | 0.57% | 1h |
| POMO | 100.00% | 100.00% | / | / | 13s | 100.00% | 100.00% | / | / | 21s |
| POMO* | 39.26% | 35.25% | 26.22 | 1.61% | 13s | 100.00% | 100.00% | / | / | 21s |
| POMO* + PIP | 5.54% | 2.67% | 25.66 | 0.18% | 15s | 31.49% | 16.27% | 51.42 | 0.37% | 48s |
| POMO* + PIP-D | 6.76% | 3.07% | 25.69 | 0.28% | 15s | 13.18% | 6.48% | 51.39 | 0.31% | 48s |

# Results are adopted from [9] due to unavailable source code, with our 'Easy' settings corresponding to their 'Medium' dataset.
/ The corresponding results are not available due to no feasible solutions or not given by [9].
† We report the average results for instances where feasible solutions were found, which vary across different models. Despite these variations, the results for overlapping feasible instances consistently show similar patterns (see Appendix D.2).

## 5 Experiments

In this paper, we propose a Proactive Infeasibility Prevention (PIP) framework and its enhanced version, PIP-D, to address the limitations of existing masking mechanisms for handling complex constraints. Notably, our PIP and PIP-D are generic and can be applied to boost various problem variants and neural methods. To evaluate the effectiveness of our method, we apply our PIP frameworks to two representative AR constructive methods, AM [4] and POMO [5], and the latest NAR constructive GFACS [10]. For the benchmark problem, we consider two representative complex VRP variants with strong interdependent constraints that challenge existing neural methods (i.e., TSPTW and TSPDL, each at varying levels of hardness) with small problem scale $n = 50, 100$ for AM [4] and POMO [5] and large scale $n = 500$ for GFACS [10]. All the experiments are conducted on servers with NVIDIA GeForce RTX 3090 GPUs and Intel(R) Xeon(R) Gold 6326 CPU at 2.90GHz. Our implementation in PyTorch are publicly available at https://github.com/jieyibi/PIP-constraint.

**Implementation details.** We generate instances at different hardness levels following prior works. For TSPTW [7, 9, 30, 67], we generate three types of instances: Easy, Medium and Hard, by adjusting the width and overlap of the time window. For TSPDL [68–70], we consider two levels of hardness: Medium and Hard. More details of such instance generation are provided in Appendix A. To ensure a

Table 2: Experiments on TSPDL instances with two different hardness.

| | Method | Infeasible% Sol.↓ | Infeasible% Inst.↓ | Obj.↓ | Gap↓ | Time↓ | Infeasible% Sol.↓ | Infeasible% Inst.↓ | Obj.↓ | Gap↓ | Time↓ |
|---|---|---|---|---|---|---|---|---|---|---|---|
| | | | | $n=50$ | | | | | $n=100$ | | |
| **Medium** | LKH3 | 0.00% | 0.00% | 10.87 | 0.00% | 5.1h | 0.00% | 0.00% | 16.39 | 0.00% | 14h |
| | ORTools | 100.00% | 100.00% | / | / | 10.9s | 100.00% | 100.00% | / | / | 56.9s |
| | Greedy-L | 100.00% | 100.00% | / | / | 2.4m | 100.00% | 100.00% | / | / | 9.5m |
| | Greedy-C | 0.00% | 0.00% | 26.09 | 144.24% | 9.1s | 0.00% | 0.00% | 52.16 | 222.71% | 27s |
| | POMO* | 17.72% | 12.52% | 10.98 | 3.80% | 6.9s | 49.39% | 32.19% | 17.11 | 9.15% | 18s |
| | POMO* + PIP | 2.21% | 0.43% | 11.22 | 3.41% | 8.5s | 2.88% | 0.38% | 17.71 | 8.08% | 31s |
| | POMO* + PIP-D | 2.64% | 0.37% | 11.26 | 3.78% | 8.4s | 2.14% | 0.23% | 17.84 | 8.86% | 31s |
| **Hard** | LKH3 | 0.00% | 0.00% | 13.30 | 0.00% | 6.8h | 0.00% | 0.00% | 20.70 | 0.00% | 1.2d |
| | ORTools | 100.00% | 100.00% | / | / | 10.6s | 100.00% | 100.00% | / | / | 56.8s |
| | Greedy-L | 100.00% | 100.00% | / | / | 2.4m | 100.00% | 100.00% | / | / | 9.4m |
| | Greedy-C | 0.00% | 0.00% | 26.07 | 99.73% | 10.9s | 0.00% | 0.00% | 52.17 | 156.37% | 25s |
| | POMO* | 37.01% | 29.25% | 13.03 | 4.11% | 6.8s | 99.98% | 99.85% | 20.95 | 15.87% | 18s |
| | POMO* + PIP | 4.53% | 2.10% | 13.66 | 3.13% | 8.5s | 28.55% | 20.66% | 22.30 | 12.67% | 31s |
| | POMO* + PIP-D | 3.89% | 0.82% | 13.80 | 3.95% | 8.5s | 12.84% | 7.91% | 22.84 | 12.32% | 31s |

comprehensive comparison, we also train and evaluate the models learned solely using our designed Lagrangian multiplier method. Meanwhile, we mark the models that use the Lagrangian multiplier with an ∗ for clarity. For our proposed approaches, our PIP models build on the Lagrangian multiplier by further incorporating one-step preventative infeasibility masking, while the enhanced version, PIP-D, is trained with a periodically and adaptively updated PIP decoder as previously described. Hyper-parameters for training follow the original settings of the backbone models except for the ones related to the added PIP decoder. Detailed hyper-parameters and additional results are available in Appendix C and D. During inference, we adhere to the settings of the original backbone models. For the AM series models, we sample 1280 solutions per instance; for the POMO series models, we use a greedy strategy with $8\times$ augmentation; and for the GFACS series models, we employ 100 ants to generate solutions for each instance over 10 pheromone iterations.

**Baselines.** We compare our proposed PIP framework with two types of baselines: 1) heuristic methods, including *LKH3* [71], a strong solver designed for multiple VRP variants; *OR-Tools* [72], a more flexible solver allowing different combinations of multiple diverse constraints; and *Greedy Heuristics* that selects locally optimal candidates at each step, where *Greedy-L* picks the nearest candidate and *Greedy-C* chooses based on complex constraints: in TSPTW, the soonest time window ends relative to the current time; and in TSPDL, the minimal draft limit; 2) Neural methods, including the original *AM* [4], *POMO* [5] and *GFACS* [10], as well as *JAMPR* [73], adapted by [9] to solve TSPTW from VRPTW; and *MUSLA* [9], a prior work on TSPTW trained in supervised manner, where *OSLA* is its one-step version and *MUSLA adapt* adopts an adaptive inference strategy. More details on the compared baselines are presented in Appendix C.

**Evaluation metrics.** In this paper, we report the following metrics to evaluate the performance of our proposed PIP framework: 1) the ratio of infeasible solutions (Infeasible%), which includes the solution-level (Sol.) infeasible rate that considers all generated solutions during inference and the instance-level (Inst.) infeasible rate that considers the comprehensive results of $N_s$ solutions generated by the sampling ($N_s = 1,280$ in AM series models) or augmentation ($N_s = 8$ in POMO series models). If at least one feasible solution is found among these $N_s$ solutions, the instance is considered to have feasible solutions; 2) average optimality gap (Gap) w.r.t the strong baseline LKH [71] for the best feasible solutions within $N_s$ solutions; 3) average tour length (Obj.) of the feasible best solutions within $N_s$ solutions; and 4) inference time, where we report the total time taken to solve 10,000 ($n = 50$ and 100) or 128 ($n = 500$) instances, with batch parallelism enabled on a single GPU. For baselines run in CPU, we exhibit the results in parallel on 16 CPU cores.

## 5.1 Model performance on complex constrained problems

The performance comparison on TSPTW and TSPDL at various levels of problem hardness is presented in Table 1 and Table 2, respectively. Notably, the original backbone models AM and POMO could not solve the problem even at the easiest level. By incorporating the Lagrangian multiplier (indicated by *), the models begin to generate some feasible solutions. However, this advantage diminishes under more complex constraints. For example, the instance-level infeasibility rates for

POMO and AM on Hard TSPTW-100 reach 100% in Table 1, which is dramatically reduced to 6.28% with the addition of PIP-D, while also improving solution quality. Compared to traditional heuristics like ORTools, Greedy-L, and Greedy-C, our PIP-D consistently outperforms these methods and shows favourable results against JAMPR and MUSLA, especially in large-scale problems. Furthermore, compared to PIP, our PIP-D delivers competitive or even better objective values and optimality gaps while significantly enhancing training efficiency (e.g., 1.5 times faster for $n = 50$ and 5.8 times faster for $n = 100$, w.r.t POMO* + PIP). Notably, the superiority of PIP-D is more significant on the more constrained hardness levels and larger problem sizes. For TSPDL, we observe similar patterns, where our PIP and PIP-D models consistently outperform other baselines in terms of both infeasibility reduction and solution quality. These results validate that our PIP approach significantly reduces infeasible rates and substantially improves solution quality compared to existing neural methods.

## 5.2 Model performance on large-scale problems

We further evaluate the capability of solving large-scale problems by implementing our PIP framework on GFACS [10]. As displayed in Table 3, equipping GFACS with our PIP significantly reduces the infeasible rate, for both the solution level and instance level and simultaneously enhances solution quality. Notably, GFACS* + PIP-D almost guarantees to obtain all feasible solutions. Different from AM and POMO, GFACS is a NAR constructive solver, which showcases the generality of our framework.

Table 3: Results on Medium TSPTW-500.

| Method | Infeasible% | | Gap↓ | Time↓ |
| | Sol.↓ | Inst.↓ | | |
| --- | --- | --- | --- | --- |
| LKH3 | 0.00% | 0.00% | 0.00% | 26m |
| Greedy-L | 100.00% | 100.00% | / | 3.2m |
| Greedy-C | 100.00% | 100.00% | / | 4.1s |
| GFACS* | 58.20% | 57.81% | 21.32% | 6.4m |
| GFACS* + PIP | 4.72% | 1.56% | 15.04% | 6.5m |
| GFACS* + PIP-D | 0.03% | 0.00% | 11.95% | 6.5m |

## 5.3 Further Experiments

**Ablation on each PIP and PIP-D design.** We now provide in-depth discussions on the effectiveness of the three proposed designs: the Lagrangian multiplier (*), the PI masking (PIP) and the learnable decoder (PIP-D). As shown in Tables 1 and 2, in *Easy* datasets, the solution-level infeasible rate for POMO* is 2.11%, improving to 0.06% with PIP-D. This shows that, for less complex constraints, the Lagrangian multiplier alone effectively guides the policy to feasible regions, hedging the impact of PIP and PIP-D, which aligns with Figure 2(b) and (d) where $\Pi_F$ is relatively large compared to $\Pi$. However, in more complex scenarios, where $\Pi_F$ is much smaller relative to $\Pi$ (as in Figure 2(c)), the neural policy struggles even with the Lagrangian multiplier. In such cases, our PIP and PIP-D become crucial, significantly confining the search space as depicted in Figure 2(e). In *Medium* datasets, the infeasible rate drops from 18.7% in POMO* to 3.34% in POMO* + PIP-D; in *Hard* datasets, it drops dramatically from 100% in POMO* to 6.48% in POMO* + PIP-D. These results verify that our PIP framework achieves significant improvement, especially as problem complexity increases.

**Ablation on the terms in Lagrangian function.** In Figure 4, we exhibit the results with and without the $\mathcal{J}_{IN}$ in Eq.(3), which validates its efficacy of enhancing the constraint awareness.

**Ablation on weighted balancing strategy.** Recall that the ratio of infeasible to feasible samples in PIP labels varies with the inherent constraint hardness. Our preliminary experiments suggest that such a ratio can reach up to 20:1 in the case of Hard datasets, causing significant label imbalance. This imbalance may significantly impact the performance of POMO* + PIP-D on several datasets, especially the harder ones, leading to 0% prediction accuracy on the minority class and causing a 100% infeasible rate for the backbone solver without a weighted balancing strategy. This indicates that the hardness-adaptive label balance strategy is essential. Moreover, for the accuracy of PIP-D, please refer to Appendix D.4.

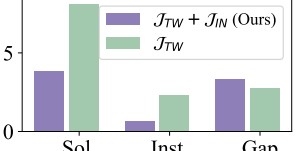

Figure 4: Effects of $\mathcal{J}_{IN}$

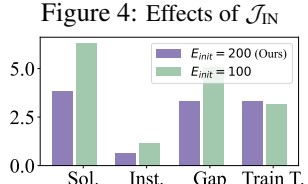

Figure 5: Effects of Less Update.

**Ablation on periodical update strategy.** In Figure 5, we evaluate PIP-D models with fewer updates. Results show that more updates improve performance, despite a slight increase in training time.

**Ablation on different step numbers.** Instead of iterating over all future possibilities, we use one-step PI masking to approximate NP-hard feasibility mask and reduce computational cost. To provide a

Table 4: Results of PIP steps on Medium TSPTW-50.

| Method | PIP Step | Sol. Infsb%↓ | Inst. Infsb%↓ | Gap↓ | Time↓ |
|---|---|---|---|---|---|
| POMO*+PIP | 0 | 76.92% | 47.28% | 4.24% | 13s |
| POMO*+PIP | 1 | 4.53% | 0.90% | 2.91% | 15s |
| POMO*+PIP | 2 | 2.90% | 0.50% | 2.93% | 4.2m |
| POMO*+PIP-D | 0 | 47.86% | 20.86% | 3.49% | 13s |
| POMO*+PIP-D | 1 | 3.83% | 0.65% | 3.32% | 15s |
| POMO*+PIP-D | 2 | 2.59% | 0.35% | 3.34% | 4.2m |

Table 5: Results of PIP steps on Hard TSPTW-100.

| Model | PIP Step | Sol. Infsb%↓ | Inst. Infsb%↓ | Gap↓ | Time↓ |
|---|---|---|---|---|---|
| POMO*+PIP | 0 | 100.00% | 100.00% | / | 21s |
| POMO*+PIP | 1 | 31.49% | 16.27% | 0.37% | 48s |
| POMO*+PIP | 2 | 26.87% | 12.88% | 0.37% | 35m |
| POMO*+PIP-D | 0 | 79.73% | 63.29% | 0.31% | 21s |
| POMO*+PIP-D | 1 | 13.18% | 6.48% | 0.31% | 48s |
| POMO*+PIP-D | 2 | 11.62% | 5.63% | 0.31% | 35m |

Table 6: Results on LKH3 with the similar instance inference time limit as POMO*+PIP(-D).

| | Method | $n = 50$ | | | $n = 100$ | | |
|---|---|---|---|---|---|---|---|
| | | Inst. Time | Obj. | Inst. Infsb% | Inst. Time | Obj. | Inst. Infsb% |
| Easy | LKH3 (Default) | 27s | 7.31 | 0.00% | 49s | 10.21 | 0.00% |
| | LKH3 | 0.37s | 7.35 | 0.00% | 0.9s | 10.37 | 0.00% |
| | POMO*+PIP | 0.38s | 7.50 | 0.00% | 0.9s | 10.57 | 0.00% |
| | POMO*+PIP-D | 0.38s | 7.49 | 0.00% | 0.9s | 10.66 | 0.00% |
| Medium | LKH3 (Default) | 40s | 13.02 | 0.00% | 1.0m | 18.74 | 0.00% |
| | LKH3 | 0.37s | 13.06 | 0.00% | 0.9s | 19.00 | 0.00% |
| | POMO*+PIP | 0.38s | 13.40 | 0.90% | 0.9s | 19.61 | 0.19% |
| | POMO*+PIP-D | 0.38s | 13.45 | 0.65% | 0.9s | 19.79 | 0.03% |
| Hard | LKH3 (Default) | 40s | 25.61 | 0.12% | 3.2m | 51.24 | 0.07% |
| | LKH3 | 0.37s | 25.43 | 30.60% | 0.9s | 49.94 | 97.28% |
| | POMO*+PIP | 0.38s | 25.66 | 2.67% | 0.9s | 51.42 | 16.27% |
| | POMO*+PIP-D | 0.38s | 25.69 | 3.07% | 0.9s | 51.39 | 6.48% |

Table 7: Results on LKH3 with the similar total inference time limit as POMO*+PIP(-D).

| | Method | $n = 50$ | | | $n = 100$ | | |
|---|---|---|---|---|---|---|---|
| | | Total Time | Obj. | Inst. Infsb% | Total Time | Obj. | Inst. Infsb% |
| Easy | LKH3 (Default) | 4.6h | 7.31 | 0.00% | 8.5h | 10.21 | 0.00% |
| | LKH3 | 26s | 8.81 | 99.29% | 58s | / | 100.00% |
| | POMO*+PIP | 21s | 7.50 | 0.00% | 48s | 10.57 | 0.00% |
| | POMO*+PIP-D | 21s | 7.49 | 0.00% | 48s | 10.66 | 0.00% |
| Medium | LKH3 (Default) | 7h | 13.02 | 0.00% | 10.8h | 18.74 | 0.00% |
| | LKH3 | 25s | 13.05 | 39.91% | 63s | / | 100.00% |
| | POMO*+PIP | 21s | 13.40 | 0.90% | 48s | 19.61 | 0.19% |
| | POMO*+PIP-D | 21s | 13.45 | 0.65% | 48s | 19.79 | 0.03% |
| Hard | LKH3 (Default) | 7h | 25.61 | 0.12% | 1.4d | 51.24 | 0.07% |
| | LKH3 | 22s | / | 100.00% | 54s | / | 100.00% |
| | POMO*+PIP | 21s | 25.66 | 2.67% | 48s | 51.42 | 16.27% |
| | POMO*+PIP-D | 21s | 25.69 | 3.07% | 48s | 51.39 | 6.48% |

comprehensive picture of the computational trade-offs, we further conduct experiments on PIP and PIP-D with different step numbers. In Table 4 and 5, we gather the results (solution feasibility and quality) and the inference time for different PIP steps. Results suggest that zero-step PIP saves time but suffers from unacceptable performance; the two-step PIP improves performance slightly but is computationally expensive. Hence, one-step PIP balances these trade-offs effectively.

**Comparison with LKH3 under different inference time budget.** To provide a more comprehensive comparison with LKH3, we provide additional results of LKH3 with identical time limits as the proposed approach across varying instance difficulty levels (Easy, Medium, Hard) and scales ($n = 50, 100$ nodes). The time limits are configured in two ways: matching the *per instance inference time* without parallelization and matching the *total inference time* with parallelization on a GPU. As shown in Table 6, POMO*+PIP(-D) outperforms LKH3 on Hard datasets, while maintaining competitive results on Easy and Medium datasets. While comparing per-instance time might seem fair for CPU-based LKH3, ignoring parallelization could disadvantage GPU-based solvers (thus not fair for GPU-based solvers). To further explore this, we conduct another experiment but with a similar total inference time limit across both methods, which is a common practice in most existing NCO papers. Results, in Table 7, show that our POMO*+PIP(-D) performs consistently better than LKH3 across most of the hardness. Moreover, to leverage the strengths of both approaches and further reveal the practical usage of our method, we explore a hybrid method that combines our PIP(-D) framework with LKH3. The results, as shown in Appendix D.3, reveal that LKH3's search efficiency can be significantly enhanced when initialized with solutions from our PIP(-D) framework.

## 6 Conclusions

In this paper, we study an unsolved challenge in neural VRP solvers and correspondingly propose a novel Proactive Infeasibility Prevention (PIP) framework to advance their capabilities towards addressing VRPs with complex constraints. Technically, we introduce a Lagrangian multiplier method and preventative infeasibility masking to proactively guide the solution construction process. By further incorporating an auxiliary decoder, our PIP framework enhances training efficiency while exhibiting superior performance on more complex datasets. While our PIP is generic and has shown great ability to boost both AR and NAR constructive methods, one potential limitation is that it may not improve performance on all backbone solvers and all VRP variants. Future directions include: 1) exploring other strategies to reduce computational complexity, such as employing a trainable heatmap to confine the candidate space of PI masking calculation, 2) applying PIP to more neural methods at larger scales, 3) extending PIP to neural iterative solvers, 4) applying PIP to more VRP variants with complex constraints, including those hard-constrained VRPs whose feasibility masking is not NP-hard but with large optimality gaps, 5) exploring the applications of PIP in other domains, such as job shop scheduling, where operations need to be completed in a specific order and infeasibility can be proactively prevented using PIP, and 6) developing theoretical justifications for PIP.

## Acknowledgments and Disclosure of Funding

This research is supported in part by the National Research Foundation, Singapore under its AI Singapore Programme (AISG Award No: AISG3-RP-2022-031) and in part by the Singapore Ministry of Education (MOE) Academic Research Fund (AcRF) Tier 1 grant. We are grateful to Dr. Yingpeng Du and Dr. Yuan Jiang for the constructive discussions. We would like to thank the anonymous reviewers and (S)ACs of NeurIPS 2024 for their constructive comments and service to the community.

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

# Learning to Handle Complex Constraints for Vehicle Routing Problems (Appendix)

## A  Details of considered VRPs

In this paper, we mainly consider two VRP variants with complex interdependent constraints: Traveling Salesman Problem with Time Window (TSPTW) and TSP with Draft Limit (TSPDL). We first detail their corresponding data generation process, then demonstrate the interdependent nature of constraints inherent in these problem variants.

### A.1  Traveling salesman problem with time window (TSPTW)

Each TSPTW instance includes a depot node and $n$ customer nodes with four properties: 2-dimensional coordinates in Euclidean space $(x_i, y_i)$, lower bound $(l_i)$ and upper bound $(u_i)$ of time windows. In prior works [7, 9, 30, 67], they generate the coordinates following a uniform distribution confined in a square box $(x_i, y_i) \sim \mathcal{U}[0, 100]$ while generating time window differently. Concretely, there are three ways to synthesize the time window: 1) construct a near-optimal TSP solution first and generate the time window according to the distance between the two adjacent nodes in the pre-generated near-optimal solution (e.g. in [74, 75]); 2) construct random node permutation first and generate time window according to the distance between the two adjacent nodes in the pre-generated random solution (e.g. in [30, 67, 76]); and 3) generate the time window under a uniform distribution without prior TSP permutation (e.g. in [7, 9]). The first two methods generate the time window based on a pre-generated TSP solution which can guarantee the existence of feasible solutions for the generated instances. However, they diminish the impact of time window constraints due to the strong prior knowledge of TSP and the pre-generated solutions. Meanwhile, the first method necessitates obtaining a near-optimal solution for TSP initially, which incurs additional computational costs. In contrast to the first two methods, the third method appears more generic but does not guarantee feasibility. Below, we describe the detailed settings for instance generation, considering three different levels of hardness, as outlined in the main paper.

**Easy TSPTW.** We mainly follow the settings from recent works [7, 9] and employ the third method presented above to generate time window. In specific, the lower bound of the time window $l_i$ follows a uniform distribution, i.e., $l_i \sim \mathcal{U}[0, T_N]$, where $T_N$ is an estimator of expected tour length in relation to the problem scale. For example, $T_{20} \approx 10.9$ [9]. The upper bound of the time window $u_i$ is generated based on $l_i$, where $u_i \sim l_i + T_N \cdot \mathcal{U}[\alpha, \beta]$, and $\alpha$ and $\beta$ are set to 0.5 and 0.75, respectively.

**Medium TSPTW.** It follows the same settings of the easy TSPTW, except that $\alpha$ and $\beta$ are set to 0.1 and 0.2, respectively. To decrease $\alpha$ and $\beta$, we derive TSPTW instances with tighter time windows, resulting in an increased hardness level.

**Hard TSPTW.** Different from easy and medium TSPTW, hard instances are generated following the settings of the benchmark dataset [67]. The second method is leveraged to generate time window. Concretely, we first obtain a permutation $\tau$ by randomly shuffling nodes. Then, the time window is generated based on $\tau$ following a uniform distribution, $l_i \sim \mathcal{U}[\psi_i - \eta, \psi_i]$ and $u_i \sim \mathcal{U}[\psi_i, \psi_i + \eta]$, where $\psi_i$ denotes the cumulative distance of the partial solution until time step $i$. $\eta$ is a factor to control the width of the time window. As preliminary experiments [9] suggest, the original $\eta$ was set at 500, allowing the instance to achieve optimality using a simple greedy heuristic. Hence, in this paper, we set $\eta$ to 50 to generate tighter time windows and increase hardness.

Following the conventions [4, 5], we normalize the node coordinates into $[0, 1]$ by dividing a scale factor $\rho = 100$. Pertaining to the time window, we also divide $l_i$ and $u_i$ by $\rho$. In specific, following [30], we first modify the upper bound of the depot $u_0$ to the maximum value of the upper bound of time window among all the customer nodes plus the travelling distance between them, i.e, $u_0 = \max(u_i + ||v_i - v_0||_2), i \in [1, n]$. Then we use $u_0$ as the normalization factor, scaling all $l_i$ and $u_i$ by it to confine their values within the range of [0, 1].

## A.2 Traveling salesman problem with draft limit (TSPDL)

TSPDL is prevalent in marine transportation scenarios, which consider the vessel capacity of the freighters. Each node has its own demand $\delta_i$ and draft limit $d_i$. A TSPDL instance is derived from a TSP instance by mutating the draft limit to less than the total demand for $\sigma\%$ of nodes, i.e., $d_i \sim [\delta_i, \sum_{j=0}^n \delta_j]$, while the draft limits of remaining nodes are equal to the total demand, i.e., $d_i = \sum_{j=0}^n \delta_j$. We can adjust $\sigma$ to manipulate the hardness of the TSPDL dataset. The demand is set to one for customer nodes or zero for depot node $v_0$ following benchmark datasets [68, 69, 71]. The mutation proportion $\sigma\%$ varies, being either a random value or fixed to 10, 25, 50, or 75 percent. We observe that a small $\sigma\%$ results in a simple TSPDL dataset that can be effectively managed with heuristics. In this paper, we focus on relatively hard problems by setting $\sigma\%$ to 75% for the *Medium* dataset and 90% for the *Hard* dataset. Note that the availability of feasible solutions can be guaranteed through the feasibility check, as demonstrated in the following PyTorch implementation.

```
node_demand = torch.cat([torch.zeros((batch_size, 1)), torch.ones((batch_size,
    problem_size - 1))], dim=1) # shape: (batch_size, problem_size)
demand_sum = node_demand.sum(dim=1).unsqueeze(1)
for i in range(batch_size):
    feasible = False
    while not feasible:
        # mutation of dl randomly occurs in w% of the nodes except the depot
        mutation = torch.randint(1, demand_sum[i].int().item(), size =
            (problem_size * sigma // 100,))
        count = torch.bincount(mutation)
        count_cumsum = torch.cumsum(count, dim=0)
        feasible = (count_cumsum <= torch.arange(0, count.size(0))).all()
```

To summarize, the feasibility guarantee based on the data generation rules described above, and its corresponding accessibility by LKH3 [71] are presented in Table 8. For example, the data generation process of the easy TSPTW dataset cannot theoretically guarantee the existence of a feasible solution for each test instance. However, LKH3 find (at least) one feasible solution for each test instance. These results showcase that *feasible solutions consistently exist for the synthetic test instances (or datasets) used in this paper, with any reported infeasibility arising solely from the method itself.*

Table 8: Feasibility guarantee based on the generation rules and the accessibility by LKH3.

| Variant | Hardness | Feasibility Guarantee | Feasibility Accessibility by LKH3 |
|---------|----------|:---------------------:|:---------------------------------:|
| TSPTW | Easy | ✗ | ✓ |
|  | Medium | ✗ | ✓ |
|  | Hard | ✓ | ✗ |
| TSPDL | Medium | ✓ | ✓ |
|  | Hard | ✓ | ✓ |

## A.3 Irreversible solution infeasibility

As mentioned in Section 4, exiting masking mechanism fails to handle problems with complex interdependent constraints (e.g., TSPTW and TSPDL), since it may cause irreversible solution infeasibility during solution construction. Here, we provide a detailed explanation of how our preventative infeasibility masking helps twist the irreversible solution infeasibility, as illustrated in Figure 6. Given a 5-node $(v_0, v_1, v_2, v_3, v_4)$ TSPTW instance with the time window $\{[0,7], [1,4], [5,7], [2,5], [4,7]\}$, and the current partial solution $v_0 \rightarrow v_1$, we derive the preventative infeasibility mask by assuming that one of the candidates $(v_2, v_3, v_4)$ is visited and check the accessibility of assumed arrival time for remaining unvisited nodes. As shown in the left panel, we assume that $v_2$ is the next visited node and see what would happen if we travel from $v_0 \rightarrow v_1 \rightarrow v_2$ to remaining unvisited nodes $v_3$ and $v_4$. We notice a constraint violation on the assumed tour $v_0 \rightarrow v_1 \rightarrow v_2 \rightarrow v_3$ since the assumed current time is already 5 at node $v_2$; thus, the earliest arrival time to $v_3$ is 7, which falls outside the time window of $v_3$. Note that another assumed tour $v_0 \rightarrow v_1 \rightarrow v_2 \rightarrow v_4$ will also violate the time window of $v_3$ in the end, since it further pushes the assumed arrival time at $v_3$. Therefore, $v_2$

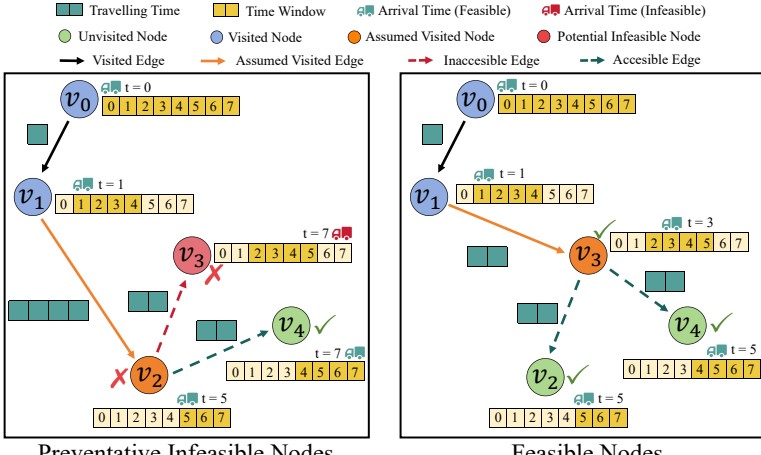

Figure 6: A TSPTW example of Preventative Infeasibility Masking in handling irreversible solution infeasibility. A dark green square denotes an unit of the travel time. No service time is considered.

is marked as an infeasible node by our preventative infeasibility masking, and then it will further consider other candidate nodes (i.e., $v_3$ and $v_4$), given the current partial solution $v_0 \rightarrow v_1$, following the same logic. Intuitively, this infeasibility is caused by the wrong selection of $v_2$ in the current step, which pushes the current time to a high value that exceeds time windows of some remaining unvisited nodes, resulting in irreversible solution infeasibility. With that said, once some candidates with late time windows have been selected, it will cause a lapse of current time, which irreversibly influences the remaining unvisited nodes (e.g., with early time windows) in future steps of solution construction. Such phenomena can also be observed in other interdependent constraints that exhibit incremental or quantified properties, such as demand-related constraints, vehicle number constraints, etc.

## B   Network architecture of PIP decoder

The network architecture of our PIP-D framework generally mirrors that of the backbone model, but it incorporates dual decoders (i.e., routing decoder and PIP decoder). For the PIP decoder, we replace the final output layer with a Sigmoid layer to adjust the output range, unless the backbone model's last layer is already a Sigmoid, as in GFACS [10], in which case no replacement is necessary.

Specifically, the AR backbone models AM [4] and POMO [5] share a similar architecture, with a multi-head attention (MHA) layer serving as the foundation. Without loss of generality, we take POMO decoder as an example for a demonstration purpose. As shown in Figure 3, the decoder receives the node embedding $h_i$, the solution embedding $h^s$, and the real-time solution feature $f^s$ as inputs, which are then used to compute the query ($q$), key ($k$), and value ($v$) for the MHA layer:

$$q^b = W_q^b[h^s, f^s], \quad k_i^b = W_k^b h_i, \quad v_i^b = W_v^b h_i, \tag{5}$$

where $W_q^b$, $W_k^b$, and $W_v^b$ are parameter matrices of the $b_{th}$ ($b \in [1, B]$) attention head, and [,] denotes the concatenation operator. The output of the MHA layer is calculated as:

$$a^b = \sum_{i=0}^{n} \text{Softmax} \left( \frac{(q^b)^T k^b}{\sqrt{d_k}} \right)_i v_i^b, \tag{6}$$

where $d_k$ is the dimension of the key. Subsequently, the output of each head passes through a linear layer parameterized by $W_o$, resulting in $h_a = W_o[a^0, a^1, \ldots, a^B]$. The decoder then computes the selection probabilities for all candidates nodes using a single-head attention layer:

$$p_i = \text{Softmax} \left( \xi \cdot \tanh \left( \frac{h_a^T h_i}{\sqrt{d_h}} \right) \right), \tag{7}$$

where $d_h$ is the dimension of the node embedding. $\xi$ is used to clip the logits to encourage policy exploration [4, 5]. For binary classification tasks in the PIP decoder, we replace the final output layer (i.e., the Softmax layer in the above equation) with a Sigmoid layer.

In contrast to AM and POMO, the encoder of GFACS outputs edge embeddings, and the decoder consists of a 3-layer multilayer perception (MLP). Each edge embedding $h_e$ is processed through:

$$h_e^{l+1} = \zeta(W_l h_e^l), \tag{8}$$

where $l \in [0, 2]$ denotes the layer index, $W_l$ represents a linear layer, and $\zeta(\cdot)$ is an activation function, i.e. SiLU for $l = 0, 1$ and Sigmoid for $l = 2$. Given the significant correlation between preventative infeasibility masking and the current partial solution $\tau_t$, we convert its NAR decoder to an AR one by adding an extra input layer that integrates the embedding of the last edge $h_{e_t}$ in $\tau_t$ and the current solution feature $f^s$ into the edge embedding itself, i.e., $h_e^0 = [h_e, h_{e_t}, f^s]$. Note that GFACS only consider $\frac{n}{5}$ (nearest) edges for each node. But for complex constrained problems (e.g., TSPTW), travelling distance may not be sufficient. Instead of using distance as the sole criterion, we select the $\frac{n}{5}$ neighbors for each node based on the extent of time window overlap, defined as $(\min(u_i, u_j) - \max(l_i, l_j))$ for all customer nodes. For the depot, since time window overlap is less relevant, we select $\frac{n}{5}$ nodes with the earliest $l_i$.

## C  Experiment details

We mark the model with Lagrangian multiplier objective function as * (e.g., AM*), and the model further with the preventative infeasibility masking as PIP (e.g., AM*+PIP). We follow their original setups (e.g., model architectures and hyper-parameters) in AM [4], POMO [5] and GFACS [10]. Pertaining to the PIP-D model (e.g., AM*+PIP-D), we employ an auxiliary decoder (i.e., PIP decoder) that is trained with the ground-truth PIP labels in a supervised manner. To balance the trade-off between training efficiency and empirical performance, we update it periodically. Specifically, within $E$ total training epochs, the PIP decoder is first trained with $E_{\text{init}}$ epochs, and then periodically updated $E_u$ epochs per $E_p$ epochs. To boost the performance, we switch $E_u$ to $E_l$ for the final $E_l$ epochs. The detailed settings are presented in Table 9. For the training epochs that utilize the outputs of the offline (i.e., freezed) PIP decoder as the preventative infeasibility masks, we employ the best-so-far PIP decoder that achieves the highest accuracy on feasible samples, since inaccurate predictions can lead to the exclusion of some feasible candidates. We set the Lagrangian multiplier $\lambda$ to 1 in the main experiments, with further analyses presented in Appendix D.1.

Table 9: Hyper-parameters of the periodical update strategy for PIP decoder.

| Method | $n$ | $E$ | $E_{\text{init}}$ | $E_p$ | $E_u$ | $E_l$ |
|---|---|---|---|---|---|---|
| POMO* + PIP-D | 50 | 10000 | 200 | 1000 | 50 | 50 |
| | 100 | 10000 | 100 | 1000 | 20 | 50 |
| AM* + PIP-D | 50, 100 | 100 | 10 | 10 | 2 | 5 |
| GFACS* + PIP-D | 500 | 50 | 10 | 10 | 2 | 5 |

**Baselines.** We compare our proposed PIP framework against the following baselines, with implementation details provided below:

- *LKH3* [71], a strong solver designed for a wide range of VRP variants, which we use to generate the (near-)optimal solutions for the test instances with 10,000 trails and 1 run.

- *OR-Tools* [72], a more flexible solver allowing different combinations of diverse constraints, which we employ the local cheapest insertion as the first solution strategy and the guided local search as the local search strategy with time limit $\mu$ for each instance (i.e., 20s for $n = 50$ and 40s for $n = 100$ following [27]). As for TSPDL, we have tried all first solution strategies outlined in the official documentation, yet we still fail to find any feasible solution.

- *Greedy Heuristics*, a classical hand-crafted method considering the local optimal candidates at each step. The *Greedy-L* heuristic selects the candidate with the shortest distance, and the *Greedy-C* heuristic selects a node based on the satisfaction of constraints, which is the candidate with the soonest time window end w.r.t current time in TSPTW and the candidate with the minimal draft limit in TSPDL.

- *AM* [4], a milestone neural AR constructive solver leveraging the Transformer architecture. We implement TSPTW and TSPDL following its default settings.

- *POMO* [5], an enhanced constructive solver upon AM by considering the symmetry property of VRP solutions. POMO shares a similar architecture with AM. We implement TSPTW and TSPDL following its default settings except removing its stipulated starting node due to the unsuitability to the complex constrained problems, which is also noted in [22].

- *GFACS* [10], a neural NAR constructive solver, which introduces the generative flow networks (GFlowNets) to improve the canonical ant colony optimization (ACO) algorithm. It incorporates a heuristic matrix and a pheromone matrix, where the former is parameterized with a neural network, and the latter is updated based on the exploration of multiple ants. We implement TSPTW on it following its default settings.

- *JAMPR* [73], a state-of-the-art model for a similar problem variant VRPTW, which was further adapted by [9] to solve TSPTW. In this paper, we directly report its result listed in [9] due to the unavailability of source code.

- *MUSLA* [9], a recent method with designs tailored for TSPTW, incorporating problem-specific features and a large supervised learning dataset, where *OSLA* is its one-step version and *MUSLA adapt* adopts an adaptive inference strategy to balance the optimality gap and the solution feasibility. Results are adopted from [9] due to the unavailability of source code, with our 'Easy' setting corresponding to their 'Medium' one in [9].

# D   Additional analyses and discussions

## D.1   Effects of different Lagrangian multiplier $\lambda$

Our PIP leverages the Lagrangian multiplier method to guide neural policy search. While some existing methods, such as [8], employ bi-level optimization techniques to jointly update the Lagrangian multiplier and the primal objective function, we find these approaches to be inefficient. Therefore, we opt to fix the value of the Lagrangian multiplier $\lambda$ and focus solely on optimizing the primal variables in this paper. Here, we investigate the effect of different Lagrangian multipliers on the solution quality and feasibility. The results are shown in the left panel of Figure 7, where we observe that a larger $\lambda$ results in a lower infeasible rate but with an increased optimality gap, whereas a smaller $\lambda$ reduces the optimality gap but at the expense of a much higher infeasible rate. Consequently, we set $\lambda = 1$ for a balance, prioritizing the optimization of the primal objective function.

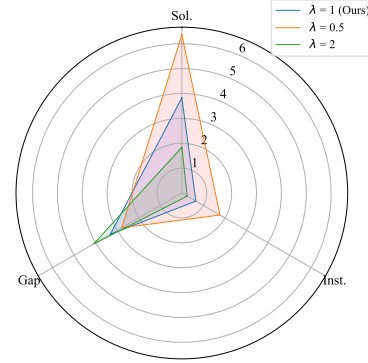

Figure 7: Radar chart of the model performance with different Lagrangian multipliers $\lambda$, including the metrics of the solution-level infeasible rate (Sol.), the instance-level infeasible rate (Inst.) and the optimality gap (Gap).

## D.2   Performance comparison on overlap feasible instances

We report the average objective values and optimality gaps in Tables 1 and 2. It should be noted, however, that these metrics are calculated across different sets of feasible instances. Considering that a complete overlap of feasible instances across all baselines is impractical (due to the 100% infeasibile rates of some baselines), we present additional results on a set of overlapped feasible instances across POMO variants to facilitate a more conprehensive comparison. As shown in Table 10, the models with our PIP consistently outperforms POMO in terms of solution quality. On the easy dataset, PIP-D exceeds PIP due to the incorporation of data augmentation during inference. In summary, despite the different instance sets, the optimality gaps displayed in Table 1 (*w/o* overlap) exhibit numerical patterns and conclusions that align with those empirically observed on the overlapped sets.

Table 10: The optimality gap of overlapped feasible instances among 10000 TSPTW-50 instances.

| | Method | Gap | Overlap Gap w. Aug. | Overlap Gap w/o Aug. |
|---|---|---|---|---|
| Easy | POMO* | 3.08% | 3.08% | 5.95% |
| | POMO* + PIP | 2.65% | 2.65% | **4.87%** |
| | POMO* + PIP-D | **2.51%** | **2.51%** | 6.51% |
| | Overlap number | | 10000 | 9768 |
| Medium | POMO* | 5.23% | 5.22% | 6.83% |
| | POMO* + PIP | **2.91%** | **2.89%** | **4.43%** |
| | POMO* + PIP-D | 3.32% | 3.28% | 4.84% |
| | Overlap number | | 9524 | 8067 |
| Hard | POMO* | 1.61% | 1.62% | 1.66% |
| | POMO* + PIP | **0.18%** | **0.18%** | **0.26%** |
| | POMO* + PIP-D | 0.28% | 0.28% | 0.42% |
| | Overlap number | | 6336 | 5758 |

## D.3 Discussion on reducing the computational complexity

As illustrated in Figure 1, the feasibility masking (i.e., $n$-step PIP) in complex constraints is NP-Hard. While iterating over all future possibilities would make PI masking complete, it is computationally inefficient. Therefore, we approximate it with one-step PI masking, whose efficiency is validated in Table 4 and Table 5. To enhance training efficiency, we use an auxiliary decoder to further approximate one-step PI masking, avoiding the need to acquire it continuously during training. Besides, we further explore some strategies for accelerating the training and inference of PIP.

- **Apply sparse strategies to refine PIP calculations.** Due to the $O(n^2)$ complexity of PIP, applying it to all the unvisited candidate nodes will be computationally expensive. For large-scale problems, we only consider top K neighbours, which is implemented on GFACS. Results in Table 3 show that GFACS*+PIP-D maintains similar training and inference times as GFACS* on TSPTW-500 (i.e., 28.3h and 6.5m vs. 28.1h and 6.4m).

- **Couple with the state-of-the-art solvers (e.g. LKH3).** Our PIP is empirically verified to be efficient due to its capability to obtain good and feasible solutions within a very short time (LKH3: 1.4d vs POMO*+PIP-D: 48s), while LKH3 can get near-optimal solutions with prolonged time. To leverage the strengths of both approaches, we use our PIP-D to provide better initial solutions for LKH3. As shown in Table 11, this combination reduces the infeasibility rate from 53.11% to 0.21% and improves the objective from 51.65 to 51.25 within only a few seconds per instance. Notably, initializing LKH3 with POMO*+PIP-D outperforms the default LKH3 setup (10,000 trials), achieving slightly better solution quality while using only 27% of the inference time (9 hours vs. 1.4 days). We also show the progress of objective value and instance-level infeasibility rate over inference times in Figures 8, 9 for clearer comparison.

- **Fine-tune Lagrangian method (*) with PI masking.** On top of the basic Lagrangian method (e.g., POMO*), PIP further employs the preventative infeasibility (PI) masking throughout the training process. We would like to note that there is another way to exploit the preventative infeasibility information and reduce the computational complexity, i.e., by leveraging the PI mask to fine-tune the pretrained Lagrangian method. The comprehensive empirical results on Medium TSPTW-50 are shown in Table 12, where the first two methods denote the PIP and Lagrangian methods applied to POMO, respectively. The results indicate that a few steps of fine-tuning the Lagrangian method using PIP masks can yield favorable improvements in the feasible rate and the optimality gap, and significantly reduce the training complexity as well.

- **Early stop of PI masking.** At each solution construction (i.e., decoding) step, our PIP leverages preventative infeasibility masking to proactively steer the policy search to (near-)feasible regions, leading to increased computational overheads as the problem sizes scale up. Here, we explore the potential of early stopping PIP, where PIP is only employed during the initial steps of solution construction. Based on our empirical observation depicted in the left panel of Figure 10, infeasibility predominately occurs in the first few steps of the entire

process. This observation reveals the possibility of merely acquiring PI masks for the initial few steps, which could improve the training efficiency. We leave it to future work.

Table 11: Results of LKH3 and POMO*+PIP-D under different time limits.

| Method | Init. Sol. | LKH3 Max Trials | Total Time | Inst. Time | Inst. Infsb% | Obj. |
|---|---|---|---|---|---|---|
| LKH3 | Default | 10000 | 1.4d | 3.2m | 0.07% | 51.24 |
| LKH3 | Default | 5000 | 22.5h | 2.2m | 0.19% | 51.24 |
| LKH3 | Default | 1000 | 4h | 23s | 2.57% | 51.26 |
| LKH3 | Default | 100 | 53m | 5.1s | 53.11% | 51.65 |
| POMO*+PIP-D | / | / | **48s** | **0.9s** | 6.48% | 51.39 |
| POMO*+PIP-D+LKH3 | POMO*+PIP-D | 100 | 53m | 5.1s | 0.21% | 51.25 |
| POMO*+PIP-D+LKH3 | POMO*+PIP-D | 1000 | 9h | 52s | **0.05%** | **51.24** |

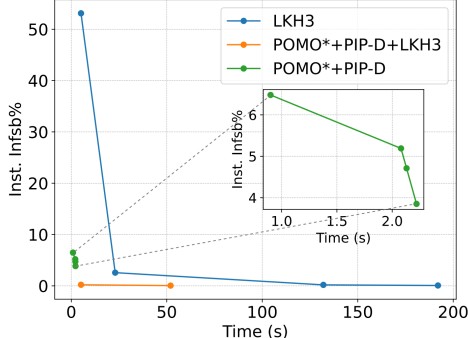 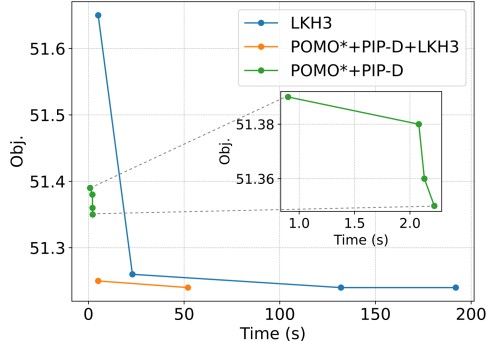

Figure 8: Average infeasibility rates over time.   Figure 9: Average objective values over time.

Table 12: Experimental results of different fine-tuning settings on Medium TSPTW-50.

| Training epoch with * | Fine-tune epochs with PIP | Training Time | Infeasible% Sol.↓ | Inst.↓ | Obj.↓ | Gap↓ |
|---|---|---|---|---|---|---|
| 10000 (POMO* + PIP) | | 120h | 4.53% | **0.90%** | 13.38 | **2.91%** |
| 10000 (POMO*) | 0 | 60h | 14.92% | 3.77% | 13.68 | 5.23% |
| 10000 | 100 | 61.3h | **4.45%** | 1.03% | 13.61 | 4.56% |
| 9900 | 100 | **60.7h** | 5.20% | 1.31% | 13.60 | 4.45% |
| 9000 | 1000 | 67h | 5.63% | 1.24% | 13.55 | 4.08% |

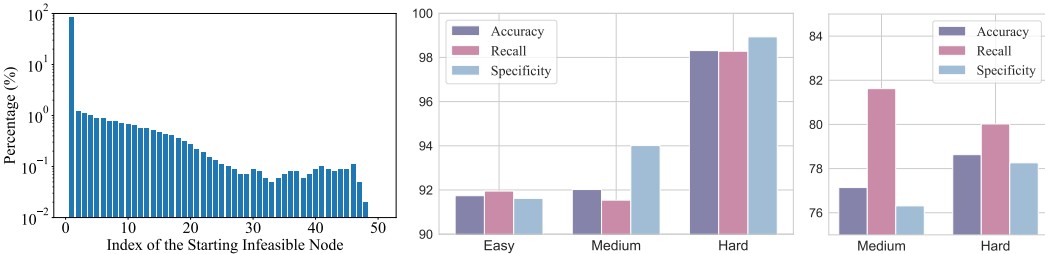

Figure 10: *Right panel:* Log-scale barplot of frequency stats of the starting infeasible node index. *Middle panel:* Evaluation metrics on TSPTW-50. *Right panel:* Evaluation metrics on TSPDL-50.

## D.4   Analyses of PIP decoder accuracy

We use an auxiliary decoder (i.e., PIP decoder) in the proposed PIP-D framework, whose goal is to learn and predict PI masks by identifying infeasible candidates based on the current partial solution. We formulate it as a binary classification task. Here, we demonstrate the efficacy of the learned PIP decoder through various evaluation metrics, including accuracy, recall (of infeasible samples), and

specificity (recall of feasible samples). As shown in the last two panels of Figure 10, our PIP decoder can accurately predict the PI masks across all hardness levels in TSPTW and TSPDL, especially for the more complex constrained one. This indicates that our PIP decoder aligns well with the goal of ensuring high accuracy in feasibility predictions.

## D.5   Performance under different inference time budget.

For a fair comparison, we also extend the inference time (by sampling more solutions and data augmentation) of the baselines to a similar one as POMO + PIP and POMO + PIP-D. Results show that incorporating the Lagrangian multiplier (POMO* and AM*) may lead to some improvement (in Table 1 and 2), but not for the cases in Table 13 under complex constraints and larger scales. Even with prolonged inference time, existing methods do not deliver any feasible solutions for the studied complex constrained VRP. In contrast, our PIP-D with 48s time significantly reduces infeasibility from 100% to 6.48% compared to baselines running for 2.5m, and exhibits an optimality gap of around only 0.3%. Furthermore, PIP can perform even better with more inference time.

Table 13: Results under different times on Hard TSPTW-100.

| Method | Inst. Infsb% | Gap | $N_s$ | Time |
|---|---|---|---|---|
| POMO (short) | 100.00% | / | 8 | 21s |
| POMO (long) | 100.00% | / | 80 | 2.5m |
| POMO* (short) | 100.00% | / | 8 | 21s |
| POMO* (long) | 100.00% | / | 80 | 2.5m |
| POMO*+PIP (short) | 16.27% | 0.37% | 8 | 48s |
| POMO*+PIP (long) | 11.75% | 0.33% | 24 | 2.4m |
| POMO*+PIP-D (short) | 6.48% | 0.31% | 8 | 48s |
| POMO*+PIP-D (long) | **5.19%** | **0.28%** | 24 | 2.4m |

## D.6   Sensitivity analyses

**Statistical significance.** To validate the statistical significance of experiments, we first conduct the Kolmogorov–Smirnov test to identify the normality of the evaluation metrics. The results indicate that the optimality gap, solution-level infeasible rate, and instance-level infeasible rate are not normally distributed. Hence, we employ the Wilcoxon test to evaluate the statistical significance. As revealed in Figure 11, the performance disparity among different methods is significant across all hardness levels, especially in more complex constrained problems, underscoring the effectiveness of our method. Note that this result further supports the improvement of our PIP framework reported in Table 1.

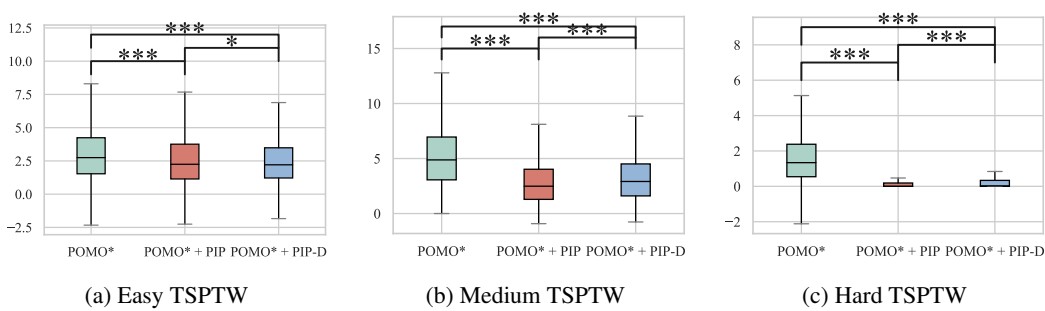

Figure 11: Boxplot of the optimality gap. In the boxplot, the * symbol denotes statistical significance, where ***, **, and * indicate significant differences between models with $p$-values $< 0.001$, $0.01$, and $0.05$, respectively, based on the Wilcoxon test.

**Performance under different hyper-parameters.** We explore the variance in model performance under different hyper-parameters. We fix the total update epoch of the PIP decoder and adjust its interval. Results in Table 14 demonstrate that our PIP model is robust to different interval settings within the same total update epoch. However, reducing the number of updates, as shown in Figure 5,

leads to a decline in model performance. Additionally, we present results using different normalization layers. Although further adjustments to the model architecture can enhance performance (e.g., using layer normalization), we adhere to the conventional settings outlined in [4] for a fair comparison.

Table 14: Sensitivity analyses of hyper-parameters on Medium TSPTW-50.

| $E_{\text{init}}$ | $E_p$ | $E_u$ | $E_l$ | Infeasible% Sol. | Inst. | Gap | Normalization | Infeasible% Sol. | Inst. | Gap |
|---|---|---|---|---|---|---|---|---|---|---|
| 200 | 1000 | 50 | 50 | 3.83% | **0.65%** | **3.32%** | Instance (Ours) | **3.83%** | 0.65% | 3.32% |
| 200 | 500 | 25 | 25 | **3.41%** | 0.68% | 3.36% | Batch | 4.14% | 0.96% | 3.10% |
| 200 | 100 | 5 | 5 | 4.60% | 1.00% | 3.34% | Layer | 4.15% | **0.58%** | **2.75%** |

## D.7 Benchmark performance

We further evaluate our PIP framework on the benchmark datasets [77] to verify the three strategies we proposed in this paper. Results show that compared to the baseline model POMO*, both our PIP and PIP-D frameworks significantly reduce the infeasibility rate and enhance solution quality.

Table 15: Model performance on the benchmark datasets [77].

| Instance | Opt. | POMO* Obj. | Gap | POMO* + PIP Obj. | Gap | POMO* + PIP-D Obj. | Gap | Instance | Opt. | POMO* Obj. | Gap | POMO* + PIP Obj. | Gap | POMO* + PIP-D Obj. | Gap |
|---|---|---|---|---|---|---|---|---|---|---|---|---|---|---|---|
| n20w20.001 | 378 | / | / | 389 | 2.91% | 389 | 2.91% | n40w40.003 | 474 | / | / | 496 | 4.64% | 497 | 4.85% |
| n20w20.002 | 286 | / | / | 292 | 2.10% | 292 | 2.10% | n40w40.004 | 452 | / | / | / | / | / | / |
| n20w20.003 | 394 | / | / | / | / | / | / | n40w40.005 | 453 | / | / | 470 | 3.75% | 471 | 3.97% |
| n20w20.004 | 396 | 405 | 2.3% | 405 | 2.27% | 405 | 2.27% | n40w60.001 | 494 | / | / | / | / | 525 | 6.28% |
| n20w20.005 | 352 | / | / | 360 | 2.27% | 360 | 2.27% | n40w60.002 | 470 | / | / | / | / | 502 | 6.81% |
| n20w40.001 | 254 | / | / | 276 | 8.66% | 279 | 9.84% | n40w60.003 | 408 | / | / | / | / | / | / |
| n20w40.002 | 333 | / | / | 347 | 4.20% | 339 | 1.80% | n40w60.004 | 382 | / | / | 406 | 6.28% | 420 | 9.95% |
| n20w40.003 | 317 | / | / | 332 | 4.73% | 332 | 4.73% | n40w60.005 | 328 | 336 | 2.4% | 342 | 4.27% | 344 | 4.88% |
| n20w40.004 | 388 | / | / | 401 | 3.35% | 401 | 3.35% | n40w80.001 | 395 | / | / | 407 | 3.04% | 407 | 3.04% |
| n20w40.005 | 288 | 314 | 9.0% | 294 | 2.08% | 302 | 4.86% | n40w80.002 | 431 | / | / | 448 | 3.94% | 452 | 4.87% |
| n20w60.001 | 335 | 377 | 12.5% | 349 | 4.18% | 353 | 5.37% | n40w80.003 | 412 | 447 | 8.5% | 444 | 7.77% | 454 | 10.19% |
| n20w60.002 | 244 | / | / | 252 | 3.28% | 260 | 6.56% | n40w80.004 | 417 | / | / | 430 | 3.12% | 435 | 4.32% |
| n20w60.003 | 352 | 369 | 4.8% | 358 | 1.70% | 358 | 1.70% | n40w80.005 | 344 | 390 | 13.4% | 362 | 5.23% | 379 | 10.17% |
| n20w60.004 | 280 | 296 | 5.7% | 298 | 6.43% | 289 | 3.21% | n60w80.001 | 458 | / | / | / | / | / | / |
| n20w60.005 | 338 | / | / | 385 | 13.91% | 361 | 6.80% | n60w80.002 | 498 | / | / | 540 | 8.43% | 548 | 10.04% |
| n20w80.001 | 329 | 348 | 5.8% | 347 | 5.47% | 347 | 5.47% | n60w80.003 | 550 | / | / | 635 | 15.45% | 646 | 17.45% |
| n20w80.002 | 338 | 390 | 15.4% | 347 | 2.66% | 360 | 6.51% | n60w80.004 | 566 | / | / | 611 | 7.95% | 632 | 11.66% |
| n20w80.003 | 320 | 361 | 12.8% | 328 | 2.50% | 328 | 2.50% | n60w80.005 | 468 | / | / | 535 | 14.32% | / | / |
| n20w80.004 | 304 | 339 | 11.5% | 341 | 12.17% | 339 | 11.51% | n80w60.001 | 554 | / | / | 582 | 5.05% | / | / |
| n20w80.005 | 264 | 312 | 18.2% | 302 | 14.39% | 302 | 14.39% | n80w60.002 | 633 | / | / | 678 | 7.11% | / | / |
| n40w20.001 | 500 | / | / | / | / | / | / | n80w60.004 | 619 | / | / | 678 | 9.53% | / | / |
| n40w20.002 | 552 | / | / | / | / | 610 | 10.51% | n80w60.005 | 575 | / | / | / | / | / | / |
| n40w20.003 | 478 | / | / | / | / | 507 | 6.07% | n80w80.001 | 624 | / | / | / | / | / | / |
| n40w20.004 | 404 | / | / | 419 | 3.71% | 418 | 3.47% | n80w80.002 | 592 | / | / | 624 | 5.41% | 638 | 7.77% |
| n40w20.005 | 499 | / | / | / | / | / | / | n80w80.003 | 589 | / | / | 648 | 10.02% | 674 | 14.43% |
| n40w40.001 | 465 | / | / | / | / | / | / | n80w80.004 | 594 | / | / | 674 | 13.47% | 676 | 13.80% |
| n40w40.002 | 461 | / | / | 485 | 5.21% | 483 | 4.77% | n80w80.005 | 570 | / | / | 627 | 10.00% | / | / |
| **Average Gap** | | 9.8% | | **5.2%** | | 5.3% | | **Average Gap** | | 8.10% | | **7.44%** | | 8.50% | |
| **Infeasible%** | | 63.0% | | 22.2% | | **14.8%** | | **Infeasible%** | | 88.89% | | **25.93%** | | 37.04% | |

## E  Broader impacts

This paper focuses on real-world scenarios and proposes a novel Proactive Infeasibility Prevention (PIP) framework to enhance the capabilities of neural methods towards solving more complex VRPs. Potential positive societal impacts include: 1) enhancing industrial efficiency, e.g., in logistics and transportation. By preemptively identifying infeasible solutions, it can reduce computational overheads and improve the efficiency of decision-making process; 2) advancing the AI and operation research (OR) communities. Our PIP framework aims to alleviate the existing challenges in the neural VRP solvers, thereby promoting the advancement of AI as well as OR. On the other hand, negative societal impacts may include environmental unfriendliness due to computational resource usage.

# F  Licenses for existing assets

The used assets in this work are listed in Table 16, which are all open-source for academic research. We will release our source code with the MIT License.

Table 16: Used assets, licenses, and their usage.

| Type | Asset | License | Usage |
|------|-------|---------|-------|
| Code | LKH3 [71] | Available for academic use | Evaluation |
| | OR-Tools [72] | Apache-2.0 license | Evaluation |
| | AM [4] | MIT License | Revision |
| | POMO [5] | MIT License | Revision |
| | GFACS [10] | MIT License | Revision |
| Datasets | Dumas et al. [77] | Available for academic use | Evaluation |

