# OpenReview forum: "Learning to Handle Complex Constraints  for Vehicle Routing Problems"
_NeurIPS.cc/2024/Conference — NeurIPS 2024 poster_

### Official Review · Reviewer_Seke · 2024-07-04

**Soundness:** 3
**Presentation:** 3
**Contribution:** 2
**Rating:** 5
**Confidence:** 4

**Summary:**

This paper introduces a novel framework called Proactive Infeasibility Prevention (PIP) to enhance the capability of neural methods in addressing complex Vehicle Routing Problems (VRPs) with interdependent constraints, such as the Traveling Salesman Problem with Time Window (TSPTW) and TSP with Draft Limit (TSPDL). The authors propose integrating the Lagrangian multiplier method and preventative infeasibility masking into the solution construction process to improve both the feasibility and optimality of solutions. Additionally, an enhanced version, PIP-D, employs an auxiliary decoder to predict infeasibility masks, potentially reducing computational costs during training. The paper presents extensive experiments demonstrating the effectiveness of PIP in reducing infeasible rates and improving solution quality.

**Strengths:**

The PIP framework represents an innovative approach to integrating constraint awareness and preventative measures within neural VRP solvers.

The use of an auxiliary decoder for predicting infeasibility masks is a good try.

The paper provides extensive empirical validation demonstrating the effectiveness of the proposed methods.

The paper is well-organized and clearly presents the problem, proposed solutions, and experimental results. The use of figures and tables effectively supports the textual content, aiding in the reader's comprehension.

**Weaknesses:**

The ideas of using the Lagrangian multiplier [1] and preventive mask functions[2] are not new.

The code is not provided, which limits the reproducibility of the work.

It's not clear how to obtain the training label of infeasibility mask.  The computation could be expensive to get a global and accurate infeasibility mask.

Some related work from recent AI and OR fields are missing.

The paper could benefit from additional experiments on a wider range of VRP variants and real-world datasets to further validate the generalizability of the proposed methods.

The presentation could be improved by providing more context on how this work relates to and advances the current state-of-the-art in neural VRP solvers.

The computational efficiency of the PIP-D framework should be more thoroughly analyzed, especially in terms of how it scales with problem size.

The paper may lack a deeper discussion on the theoretical underpinnings of the proposed methods and their potential implications for the broader field of combinatorial optimization.

[1] Qiaoyue Tang, Yangzhe Kong, Lemeng Pan, and Choonmeng Lee. Learning to solve soft-constrained vehicle routing problems with lagrangian relaxation. arXiv preprint arXiv:2207.09860, 2022.

[2] Hou, Qingchun, et al. Generalize learned heuristics to solve large-scale vehicle routing problems in real-time. The Eleventh International Conference on Learning Representations. 2023.

**Questions:**

How does the PIP framework generalize to other VRP variants beyond TSPTW and TSPDL?

Can the authors provide more insights into the computational efficiency of PIP-D, especially regarding its scalability?

**Limitations:**

The authors have acknowledged some limitations, such as the potential inability of PIP to improve performance on all backbone solvers and VRP variants. However, the paper could benefit from a more detailed discussion on the robustness of the PIP framework when faced with different levels of constraint hardness in real-world scenarios.

---

> ### Author Rebuttal · Authors · 2024-08-07
>
> We thank the reviewer for constructive comments and acknowledging that our PIP is innovative, and extensively validated, and our paper is well-organized and clear. We understand the need for more discussion on related works, computational costs, and code release. We hope the response below and the added new experiments would clarify any misunderstandings and concerns.
>
> ---
> **[Discussion on [1][2]]** Thank you for pointing out the references. We have reviewed them carefully and would like to clarify that our proposed PIP is mostly distinct from works [1] and [2], which correspond to [8] and [51] in the original paper, respectively.
> - **Regarding [1]:** While both are early attempts to incorporate Lagrangian multiplier into NCO, [1] is designed for **iterative solvers for soft-constrained VRPs**, whereas PIP is for **constructive solvers for hard-constrained VRPs**. Moreover, our primary motivation for using the Lagrangian multiplier is to enhance our model's initial constraint awareness. Unlike [1], we argue and show that using Lagrangian multiplier alone is less effective for complex constraints. To address this, we propose our novel PIP and PIP-D, which effectively overcome this weakness and improve overall performance.
> - **Regarding [2]**: While both use masking to prevent infeasible solutions, our preventive infeasibility (PI) masking differs from the "global mask function" in [2]. Firstly, although [2] shows the effectiveness of their global masking for a specific case of CVRP with limited vehicles, our PI masking is technically different and serves as a more generic framework to address complex VRP variants where feasibility masking is NP-hard. Also, we employ an auxiliary decoder to predict PI masking, and our PIP has been validated on complex constrained variants like TSPTW and TSPDL, as well as on various autoregressive and non-autoregressive neural solvers such as AM, POMO, and GFACS, demonstrating broader applicability. Moreover, [2] primarily focuses on scalability, whereas our work emphasizes constraint handling. We note that our work can be coupled with [2] in future work, and we will discuss more in the revised paper.
> - Lastly, as acknowledged by Reviewer #DEtT, **we are among the first to address the complex interdependent constraints that make masking NP-hard**, a point not covered in existing NCO works like [1][2].
>
> ---
> **[Code Release]** As promised in line 276, we will make our code, pre-trained models, and the used data publicly available on GitHub. Following the rebuttal guidelines, **we have forwarded our code to the Area Chair.**
>
> ---
> **[How to obtain the training label]** Yes, getting global and accurate masks is expensive (as mentioned in lines 139-150). Hence, in our PIP, we employ a one-step approximation to balance computational costs without iterating over all future possibilities which is NP-hard (see results on such balance in Table R3 and R4 of the attached PDF). **The training label is such one-step masking**, determined by evaluating whether selecting a candidate node would lead to irreversible future infeasibility for the remaining unvisited nodes in the next step. If so, the candidate is considered infeasible at that step. Detailed examples and descriptions are available in lines 192-195 and Appendix A.3.
>
> ---
> **[More literature review]** We have included most recent work from NCO field and will add more from OR field. Any suggestions are warmly welcome!
>
> ---
> **[Additional experiments]**
> - Regarding real-world datasets, we have evaluated our PIP(-D) to unseen real-world instances in benchmark dataset [76] with different scales, distributions and constraint hardness in Appendix D.6 (Table 7). Additionally, we conducted extensive experiments on complex variants like TSPTW and TSPDL, covering various levels of constraint hardness (Easy, Medium, Hard). Results suggest that our PIP(-D) can generalize to real-world datasets with unknown constraint hardness.
> - Regarding other VRP variants, we follow your suggestion and explore the application of our PIP framework (Lagrangian multiplier, PI masking and the auxiliary decoder) to VRPs with various complex constraints, such as VRPBLTW (VRP with constraints of capacity, backhaul, duration limit, and time window). We found that in such variants PI masking is less important since we can obtain the masking easily at each constructive step, but the Lagrangian multiplier and the auxiliary decoder still have effects. The results demonstrate that **our PIP framework significantly enhances solution quality on VRPBLTW-50** (Our gap 1.80% vs. POMO's gap 9.17%, see Figure R1 for details).
> - Lastly, our PIP has potential beyond VRPs, such as in job shop scheduling, where operations require a specific order. PIP can proactively prevent infeasibility in these cases, and we leave them as future work.
>
> ---
> **[How our PIP advances the SOTA]** Firstly, our PIP is generic to boost an existing SOTA constructive neural VRP solver to enhance its capability of constraint handling. We have verified our effectiveness on both autoregressive (AM, POMO) and non-autoregressive (GFACS) solvers. Additionally, we have conducted newexperiments by further coupling our PIP with LKH3 (see `General Response #2` for more details).
>
> ---
> **[Computational efficiency]** Please kindly refer to our `General Response #1`.
>
> ---
> **[Theoretical analysis]** We acknowledge that the NCO domain primarily relies on empirical results and often lacks theoretical support for most of published papers. Our findings are backed by extensive results, demonstrating empirical superiority with significant improvements in infeasibility rates and solution quality. Nevertheless, we provide some more rigour analysis for our PIP, please refer to the second response to Reviewer #azSu.
>
> ---
> **[Robustness under different hardness]** Thanks for the suggestion. We have discussed the mentioned experiments in benchmark datasets in Appendix D.6 (Table 7). We will add more analysis.

---

> > ### Author Response · Authors · 2024-08-12
> >
> > Dear Reviewer Seke,
> >
> > Thank you once again for your insightful comments and helpful suggestions. As the deadline for author-reviewer discussions is approaching, we would greatly appreciate it if you could take a moment to review our rebuttal. Please let us know if you have any further questions or concerns. Thank you very much for your time.
> >
> > Best,
> > Authors

---

> > > ### Author Response · Authors · 2024-08-14
> > >
> > > Dear Reviewer Seke,
> > > ####
> > > The discussion period is approaching its end (in less than 12 hours). Please kindly let us know if our response resolves your concerns. Regarding your concern of our code release, we have also forwarded our code to the Area Chair. We would greatly appreciate it if you could give us any feedback.
> > > ####
> > > Thank you again for your valuable comments and suggestions.
> > > ####
> > > Best Regards,
> > >
> > > Authors

---

### Official Review · Reviewer_Xb4p · 2024-07-08

**Soundness:** 3
**Presentation:** 3
**Contribution:** 2
**Rating:** 5
**Confidence:** 4

**Summary:**

This paper addresses the challenge of predicting feasible solutions for VRPs with complex constraints. It introduces two novel methods to enhance existing algorithms: i) integrating constraints directly into the optimization objective using the Lagrangian Multiplier Approach; ii) PIP framework, which proactively excludes potential nodes during solution prediction to prevent future infeasibility. Besides, a neural decoder PIP-D is proposed to reduce computational complexities.
Experiments on two VRP variants demonstrates significant advancements in feasible prediction compared to the baselines.

**Strengths:**

1. The paper is well-structured with a logical flow.
2. The feasibility issue investigated are critical, and the proposed PIP framework address it well.
3. Extensive experiments and detailed clarifications are made to demonstrate effectiveness.

**Weaknesses:**

Major concerns:

- **Computational cost.** From the numerical results, PIP incurs approximately twice the computational time compared to baseline neural solvers (e.g., AM and POMO). Despite attempts to mitigate this by introducing neural decoder PIP-D, the reported results don't exhibit a corresponding reduction in time. The authors may clarify this more.

- **Lack of comparisons** between different step numbers in PIP. The authors only conducted experiments of one-step PIP. It is crucial to include comparisons with different step numbers to assess the impact of the "looking ahead" masking strategy. For instance, additional results from zero-step PIP (directly masking nodes violating constraints) and two-step PIP. Here, I emphasize zero-step PIP since its masking strategy differs from the ones of the baseline neural solvers, as I understand it.

- Only two TSP datasets are considered. The paper is titled "Learning to Handle Complex Constraints for **Vehicle Routing Problems (VRPs)**", however both datasets are **variants of TSPs**. The key difference between VRPs and TSPs is **capacity constraints**. I would like to see VRPs with various complex constraints (such as time windows, pick-up and delivery, split delivery etc.), otherwise, the authors should change the title and consider more TSP variants.

- Heuristic algorithms for TSPs and VRPs with complex constraints have been well developed (such as LKH3). I do not see that the proposed algorithm outperforms the state-of-the-art heuristic algorithms (not neural algorithms).

**Questions:**

- What's the time cost of PIP? Is it $time(POMO^*+PIP)$ - $time(POMO^*)$?

- What are the masking strategies of baselines AM and POMO? Do they only mask visited nodes, ignoring other constraints?

- What is the stopping criteria for each algorithm? I know that running LKH3 for 1 second would often generate almost optimal solutions to TSPs with 100 customers, hence there is no need to run it for 14 hours, as shown in Table 2. I recommend the authors use the same time limit (and usually a few seconds for small-sized TSPs considered in this paper) for each baseline algorithm and even present the progress of the objective value for the best solutions found by each algorithm. This could ensure a fair comparison among them.

**Limitations:**

The author has discussed the limitations in section 6: one potential limitation is that it may not improve performance on all backbone solvers and all VRP variants.

---

> ### Author Rebuttal · Authors · 2024-08-07
>
> We appreciate the reviewer’s kind effort in providing insightful and detailed feedback. We are delighted that the reviewer finds our work to be novel, well-structured, and effective in addressing feasibility issues. We have conducted additional experiments to address your comments and hope the following response and results will clear any remaining concerns.
>
> ---
> **[Computational cost (Q1, W1)]** Thank you for your valuable comment! We apologize for any confusion. Please refer to our `General Response #1` for detailed discussion on computational costs. In summary, while our PIP introduces some unavoidable overhead to the backbone model, this is offset by its strong effectiveness. As shown by our additional results in Table R1 of the attached PDF, **even with extended inference times, existing baseline methods still struggle with constraint handling and may fail to find any feasible solutions**. In addition, we have employed strategies like one-step PI masking, an auxiliary PIP-D decoder, and a sparse strategy to further mitigate the costs. We now provide more responses:
> - **PIP does not always double the computational time of the backbone solvers.** For instance, on small-scale TSPTW-50, POMO* takes 13s while POMO*+PIP takes a similar time of 15s; on larger-scale TSPTW-500, we can leverage the sparse strategy to reduce the overhead - as shown in Table 3, our GFACS*+(PIP; PIP-D) demonstrates similar inference times to GFACS* (6.5m vs. 6.4m), while significantly reducing infeasibility (57.81% vs. 1.56%; 0.00%) and improving solution quality (21.32% vs. 15.04%; 11.95%), showing great efficiency.
> - **The auxiliary decoder aims to approximate the one-step PI masking to avoid its frequent calculation during training**. Results show that PIP-D significantly reduces the training time (e.g. by 1.5x on TSPTW-50) compared to PIP. As the inference overhead is rather insignificant than other baselines, we employ the complete one-step PI masking to guide the construction process.
> - **For Q1 - Yes, $T_{PIP}\\approx T_{POMO∗+PIP}- T_ {POMO∗}$, which results from the acquisition of the PI masking.** Detailed examples and descriptions are available in lines 192-195 and Appendix A.3. And please kindly see our next response for discussions of computational time balance.
>
> ---
> **[PIP with different step numbers (Q2, W2)]** Insightful comment! Yes, zero-step PIP differs from the baselines. Following the suggestion of the reviewer, we have further supplemented this by conducting new experiments on PIP and PIP-D with different step numbers. In Table R3, R4 attached in the PDF under `General Response`, we gather the results (solution feasibility and quality) and the inference time for different PIP steps. Results suggest that zero-step PIP saves some time but suffers from inferior performance, while the two-step PIP improves performance slightly but is computationally expensive. However, **one-step PIP balances these trade-offs effectively**. We will further clarify this in the revised paper.
>
> ---
> **[Apply PIP to more VRPs with various complex constraints (W3)]** Thank you for the suggestion. We agree that the title "Travelling Salesman Problems" might be more appropriate for our original paper. Nonetheless, we plan to follow the reviewer's suggestions and further explore applying our PIP framework to VRPs with more complex constraints, such as VRPBLTW (with capacity, backhaul, duration limit, and time window constraints). Results on VRPBLTW-50, presented in Figure R1 of the PDF under `General Response`, show that our PIP framework significantly enhances solution quality for VRPs with complex constraints (our gap 1.80% vs. POMO's gap 9.17%). While our PIP framework may not be a silver bullet for improving performance across all VRP variants, it has been successfully applied to variants like TSPTW and TSPDL, covering various levels of constraint hardness (Easy, Medium, Hard). Moreover, we believe that our PIP framework has potential applications beyond VRPs, such as in job shop scheduling, where operations need to be completed in a specific order and infeasibility can be proactively prevented. We plan to explore these applications as future work and will discuss them further in the revised paper.
>
> ---
> **[Comparison with LKH3 with time limit (Q3, W4)]** Thanks for the insightful comment. We follow your constructive suggestion and add Table R2, Figures R2 and R3 (presenting the progress of the objective value and infeasible rate over different inference time) in the attached PDF under global response. Details are discussed in our `General Response #2`.
>
> To sum up, we agree with the reviewers that LKH3 can generate near-optimal solutions for TSP-100 within a few seconds. However, **when given a limited budget, LKH3 performs significantly worse than our POMO\*+PIP-D on the studied complex variants.**  As shown in Table R2, while LKH3 is powerful with state-of-the-art quality, its advantage diminishes with limited time. Our POMO*+PIP-D reduces the infeasibility rate from 53.11% to 6.28% with 0.9s time per instance, compared to LKH3 with 5.1s time. To leverage the strengths of both approaches, we conducted an additional experiment using our PIP-D to provide better initial solutions for LKH3. This further reduces the infeasibility rate from 53.11% to 0.21% and improves the objective from 51.65 to 51.25 within only a few seconds. Notably, initializing LKH3 with POMO*+PIP-D outperforms the default LKH3 setup (10,000 trials), achieving slightly better solution quality while using only 27% of the inference time (9 hours vs. 1.4 days).
>
> Regarding the stopping criteria for each algorithm (i.e., to restrain its runtime):
> - For LKH3, we set a fixed number of max trials following the conventions in NCO;
> - For neural algorithms, we
>   - sample a pre-set number of solutions ($N_s$) for each instance: $N_s=8$ for POMO series models and $N_s=1280$ AM series models;
>   - predefine 100 ants and 10 pheromone iterations for GFACS.

---

> > ### Comment · Reviewer_Xb4p · 2024-08-08
> >
> > Thanks for addressing my comments. I still think the comparison between the proposed approach and LKH3 is not comprehensive enough (e.g., easy/medium/hard instances with more than 100 nodes, and the 500-customer instances in Section 5.2; **with the same time limit as the proposed approach**) since **LKH3 is the most important baseline algorithm** in my opinion. For example, from Figure R3, we may conclude that, for those hard instances, the proposed approach could obtain high-quality solutions quite fast but LKH3 could obtain solutions of better quality with a relatively longer time. But from Section 5.2, if we run LKH3 for a few minutes (say 6.5 min), we may already achieve a very small gap (e.g., smaller than 11.95%). That is why I said the comparison in the previous manuscript is not fair.

---

> > > ### Author Response · Authors · 2024-08-10
> > > **Thanks for the comments and please kindly check our further clarifications.**
> > >
> > > Thank you for the discussion and insightful comments!
> > >
> > > We are pleased to have addressed most of your concerns. Regarding the remaining concern about a comprehensive comparison between our PIP and LKH3, we have provided results on Hard TSPTW-100 in the previous rebuttal given the constraints of the rebuttal period and PDF page limit. Nevertheless, we understand that this may not be comprehensive enough. Following your suggestion, we now conducted additional experiments using LKH3 with the same time limits as our proposed approach **across various instance hardness levels (Easy/Medium/Hard) and scales (50/100/500 nodes)**. The time limits were set in two ways for comprehensive evaluation: similar $\underline{\text {instance inference time}}$ without parallelization (new Tables R5, R7, and R9) and similar $\underline{\text {total inference time}}$ with parallelization (new Tables R6 and R8). We hope these new empirical results offer a more comprehensive picture.
> > >
> > > **[A. For $n=50$ and $100$]**  Firstly, we follow your suggestion and only allow LKH3 to run a similar inference time as our approach $\underline{\text {per instance}}$. As shown in **Table R5**, POMO*+PIP(-D) outperforms LKH3 on Hard datasets, while maintaining competitive results on Easy and Medium datasets. While comparing per-instance time might seem fair for CPU-based LKH3, ignoring parallelization could disadvantage GPU-based solvers (thus not fair for GPU-based solvers). To further explore this, we conducted another experiment but with the similar $\underline{\text {total inference time}}$ limit across both methods. Results, in **Table R6**, show that our POMO*+PIP(-D) performs consistently better than LKH3 across most of the hardness. We recognize the challenge of achieving absolute fairness when comparing CPU-based and GPU-based solvers, and we note that total time comparison is a common practice in most existing NCO papers (e.g. those in [1-8]). We appreciate the reviewer for providing such an insightful comment, which highlights the need for developing new metrics that facilitate fairer comparisons between neural and traditional solvers—a promising direction for future work.
> > >
> > > Moreover, we agree with the reviewer that LKH3 is indeed a robust and well-developed solver. However, neural solvers offer unique advantages, such as highly efficient parallelization and reduced reliance on hand-crafted heuristic rules [9]. To leverage the strengths of both approaches and further reveal the practical usage of our method, we explored a hybrid method that combines our PIP(-D) framework with LKH3. Similarly, we provide the comprehensive experimental results conducted across various instance hardness levels (Easy/Medium/Hard) and scales (50/100/500 nodes). The results, as shown in **Table R7 and R8**, reveal that **LKH3’s search efficiency can be significantly enhanced when initialized with solutions from our PIP(-D) framework**.
> > >
> > > **[B. For $n=500$]** As shown in **Table R9**, we acknowledge that our PIP(-D) implemented on GFACS* currently underperforms LKH3. However, the integration of GFACS*+PIP(-D) with LKH3 has already made significant strides in closing this performance gap, particularly in enhancing feasibility handling compared to using the GFACS backbone alone. We would like to clarify that the primary motivation for these experiments here is to demonstrate the scalability and broad applicability of our PIP framework in enhancing various backbone neural models. While we acknowledge that PIP may not elevate every model to state-of-the-art performance, which is both reasonable due to _“No Free Lunch”_, our goal is to show its potential to significantly boost performance across a wide range of neural models.

---

> > > > ### Author Response · Authors · 2024-08-10
> > > > **Thanks for the comments and please kindly check our further clarifications (continued)**
> > > >
> > > > Additionally, we would like to clarify that:
> > > >
> > > > 1. **The motivation of this work is to highlight the issue of “NP-hard masking” in neural solvers and enhance neural solvers with the ability to handle complex constraints**, as comprehensively demonstrated in our main paper (Tables 1, 2, and 3). Thus, outperforming LKH3 is not our primary objective or key contribution. Instead, our proposed framework has significantly reduced the infeasibility rate and substantially improved solution quality for both autoregressive and non-autoregressive neural solvers. These advancements establish our framework as the new state-of-the-art in the domain of neural solvers.
> > > >
> > > > 2. **Most existing NCO solvers [2-8] use LKH3 with the default settings from [1]** (including maximum trials, number of runs, and parallelism settings) to estimate the optimality gap and provide a runtime reference. The primary focus is on understanding the performance gap between neural solvers [10, 11] and traditional solvers, rather than strictly outperforming it.
> > > >
> > > > 3. We acknowledge that **neural solvers have yet to fully replace traditional solvers like LKH3, but rather catching up with it step by step**. We believe that we provide insights for complex constraints handling for NCO, which is a step towards practical applications. Also, we would like to emphasize that our PIP(-D) is a **generic framework**, making our research orthogonal to the development of new, powerful neural solvers that have the potential to fully outperform LKH in future works.
> > > >
> > > > We appreciate the reviewer for helping our work to be more rigorous. The new results will be incorporated into the revised manuscript, along with a deeper discussion of these points. Due to the rebuttal limit, we will plot the progress curves (like Figure R2 and R3 in the attached PDF) based on the results of all the new tables for a more intuitive illustration. Also, **our code has already been forwarded to the AC**. We hope our discussions are now comprehensive. Please let us know if you have any remaining concerns.
> > > >
> > > > ####
> > > > ####
> > > >
> > > > ```
> > > > References:
> > > > [1] Attention, learn to solve routing problems!. ICLR, 2018.
> > > > [2] Reinforcement learning for solving the vehicle routing problem. NeurIPS, 2018.
> > > > [3] POMO: Policy optimization with multiple optima for reinforcement learning. NeurIPS, 2022.
> > > > [4] Simulation-guided beam search for neural combinatorial optimization. NeurIPS, 2022.
> > > > [5] Learning to search feasible and infeasible regions of routing problems with flexible neural k-opt. NeurIPS, 2023.
> > > > [6] BQ-NCO: Bisimulation quotienting for generalizable neural combinatorial optimization. NeurIPS, 2023.
> > > > [7] Neural combinatorial optimization with heavy decoder: Toward large scale generalization. NeurIPS, 2023.
> > > > [8] MVMoE: Multi-Task Vehicle Routing Solver with Mixture-of-Experts. ICLM, 2024.
> > > > [9] Machine learning for combinatorial optimization: a methodological tour d’horizon. European Journal of Operational Research, 2021.
> > > > [10] Graph-based diffusion solvers for combinatorial optimization. NeurIPS, 2023.
> > > > [11] MOCO: A Learnable Meta Optimizer for Combinatorial Optimization. arXiv preprint, 2024.
> > > > ```
> > > >
> > > > ####
> > > >
> > > > Note for Tables R7 & R8: For a fair comparison, we kept the POMO*+PIP(-D) models (with inference time denoted as $T_{PIP}$) unchanged and allowed LKH3 to search for a similar duration as $T_{PIP}$ to showcase the performance of POMO*+PIP(-D)+LKH3 (i.e., the total inference time is $2 \times T_{PIP}$). LKH3 was also given the same time limit for consistency.

---

> > > > ### Comment · Reviewer_Xb4p · 2024-08-10
> > > >
> > > > Thanks for addressing my comments. Now I'm satisfied with the comparison and raise my score. I expect the authors to include those results in the updated manuscript and edit part of their conclusions (the proposed approach vs LKH).

---

> > > > > ### Author Response · Authors · 2024-08-10
> > > > > **Thank you for your support and we are happy to have addressed your concerns!**
> > > > >
> > > > > We really appreciate your insightful suggestions and we will include the additional results in the revised paper. Thank you!

---

> ### Author Response · Authors · 2024-08-10
> **Further results for comprehensive comparison (1/4)**
>
> **Table R5**: Results on LKH3 with the similar $\underline{\text {instance inference time}}$ limit as POMO*+PIP(-D).
>
>
> |              |                |                                 |  $n=50$  |                  |                                 | $n=100$  |                  |
> |:------------:|:--------------:|:-------------------------------:|:--------:|:----------------:|:-------------------------------:|:--------:|:----------------:|
> | **Hardness** |   **Method**   | $\underline{\text{Inst. Time}}$ | **Obj.** | **Inst. Infsb%** | $\underline{\text{Inst. Time}}$ | **Obj.** | **Inst. Infsb%** |
> |     Easy     | LKH3 (Default) |               27s               |   7.31   |      0.00%       |               49s               |  10.21   |      0.00%       |
> |     Easy     |     LKH3       |              0.37s              |   7.35   |      0.00%       |              0.9s               |  10.37   |      0.00%       |
> |     Easy     |   POMO*+PIP    |              0.38s              |   7.50   |      0.00%       |              0.9s               |  10.57   |      0.00%       |
> |     Easy     |  POMO*+PIP-D   |              0.38s              |   7.49   |      0.00%       |              0.9s               |  10.66   |      0.00%       |
> |              |                |                                 |          |                  |                                 |          |                  |
> |    Medium    | LKH3 (Default) |               40s               |  13.02   |      0.00%       |              1.0m               |  18.74   |      0.00%       |
> |    Medium    |      LKH3      |              0.37s              |  13.06   |      0.00%       |              0.9s               |  19.00   |      0.00%       |
> |    Medium    |   POMO*+PIP    |              0.38s              |  13.40   |      0.90%       |              0.9s               |  19.61   |      0.19%       |
> |    Medium    |  POMO*+PIP-D   |              0.38s              |  13.45   |      0.65%       |              0.9s               |  19.79   |      0.03%       |
> |              |                |                                 |          |                  |                                 |          |                  |
> |     Hard     | LKH3 (Default) |               40s               |  25.61   |      0.12%       |              3.2m               |  51.24   |      0.07%       |
> |     Hard     |      LKH3      |              0.37s              |  25.43   |      30.60%      |             0.9s                |  49.94   |       97.28%     |
> |     Hard     |   POMO*+PIP    |              0.38s              |  25.66   |      2.67%       |              0.9s               |   51.42  |      16.27%      |
> |     Hard     |  POMO*+PIP-D   |              0.38s              |  25.69   |      3.07%       |              0.9s               |  51.39   |      6.48%       |

---

> ### Author Response · Authors · 2024-08-10
> **Further results for comprehensive comparison (2/4)**
>
> **Table R6**: Results on LKH3 with the similar $\underline{\text {total inference time}}$ limit as POMO*+PIP(-D).
>
>
> |              |                |                                 |  $n=50$  |                  |                                 | $n=100$  |                  |
> |:------------:|:--------------:|:-------------------------------:|:--------:|:----------------:|:-------------------------------:|:--------:|:----------------:|
> | **Hardness** |   **Method**   | $\underline{\text{Total Time}}$ | **Obj.** | **Inst. Infsb%** | $\underline{\text{Total Time}}$ | **Obj.** | **Inst. Infsb%** |
> |     Easy     | LKH3 (Default) |              4.6h               |   7.31   |      0.00%       |              8.5h               |  10.21   |      0.00%       |
> |     Easy     |      LKH3      |               26s               |   8.81   |      99.29%      |               58s               |    /     |     100.00%      |
> |     Easy     |   POMO*+PIP    |               21s               |   7.50   |      0.00%       |               48s               |  10.57   |      0.00%       |
> |     Easy     |  POMO*+PIP-D   |               21s               |   7.49   |      0.00%       |               48s               |  10.66   |      0.00%       |
> |              |                |                                 |          |                  |                                 |          |                  |
> |    Medium    | LKH3 (Default) |               7h                |  13.02   |      0.00%       |              10.8h              |  18.74   |      0.00%       |
> |    Medium    |      LKH3      |               25s               |  13.05   |      39.91%      |               63s               |    /     |     100.00%      |
> |    Medium    |   POMO*+PIP    |               21s               |  13.40   |      0.90%       |               48s               |  19.61   |      0.19%       |
> |    Medium    |  POMO*+PIP-D   |               21s               |  13.45   |      0.65%       |               48s               |  19.79   |      0.03%       |
> |              |                |                                 |          |                  |                                 |          |                  |
> |     Hard     | LKH3 (Default) |               7h                |  25.61   |      0.12%       |              1.4d               |  51.24   |      0.07%       |
> |     Hard     |      LKH3      |               22s               |    /     |     100.00%      |               54s               |    /     |     100.00%      |
> |     Hard     |   POMO*+PIP    |               21s               |  25.66   |      2.67%       |               48s               |  51.42   |      16.27%      |
> |     Hard     |  POMO*+PIP-D   |               21s               |  25.69   |      3.07%       |               48s               |  51.39   |      6.48%       |

---

> ### Author Response · Authors · 2024-08-10
> **Further results for comprehensive comparison (3/4)**
>
> **Table R7**: Results on POMO*+PIP(-D)+LKH3 with the similar $\underline{\text {instance inference time}}$ limit as doubled time used in POMO*+PIP(-D).
>
> |              |                  |                                 |  $n=50$  |                  |                                 | $n=100$  |                  |
> |:------------:|:----------------:|:-------------------------------:|:--------:|:----------------:|:-------------------------------:|:--------:|:----------------:|
> | **Hardness** |    **Method**    | $\underline{\text{Inst. Time}}$ | **Obj.** | **Inst. Infsb%** | $\underline{\text{Inst. Time}}$ | **Obj.** | **Inst. Infsb%** |
> |     Easy     |       LKH3       |              0.77s              |   7.33   |      0.00%       |              1.8s               |  10.31   |      0.00%       |
> |     Easy     |  POMO*+PIP+LKH3  |              0.77s              |   7.33   |      0.00%       |              1.8s               |  10.29   |      0.00%       |
> |     Easy     | POMO*+PIP-D+LKH3 |              0.77s              |   7.33   |      0.00%       |              1.8s               |  10.31   |      0.00%       |
> |              |                  |                                 |          |                  |                                 |          |                  |
> |    Medium    |       LKH3       |              0.77s              |  13.04   |      0.00%       |              1.8s               |  18.89   |      0.00%       |
> |    Medium    |  POMO*+PIP+LKH3  |              0.77s              |  13.05   |      0.00%       |              1.8s               |  18.91   |      0.00%       |
> |    Medium    | POMO*+PIP-D+LKH3 |              0.77s              |  13.05   |      0.00%       |              1.8s               |  18.92   |      0.00%       |
> |              |                  |                                 |          |                  |                                 |          |                  |
> |     Hard     |       LKH3       |              0.76s              |  25.56  |      8.81%       |              1.8s               |  50.66   |     75.54%       |
> |     Hard     |  POMO*+PIP+LKH3  |              0.76s              |  25.61   |      0.12%       |              1.8s               |  51.24   |      1.63%       |
> |     Hard     | POMO*+PIP-D+LKH3 |              0.76s              |  25.61   |      0.05%       |              1.8s               |  51.25   |      0.36%       |

---

> ### Author Response · Authors · 2024-08-10
> **Further results for comprehensive comparison (4/4)**
>
> **Table R8**: Results on POMO*+PIP(-D)+LKH3 with the similar $\underline{\text {total inference time}}$ limit as doubled time used in POMO*+PIP(-D).
>
> |              |                  |           |                                 |    $n=50$     |                  |              |                                 | $n=100$   |                  |
> |:------------:|:----------------:|:---------:|:-------------------------------:|:-------------:|:----------------:|:------------:|:-------------------------------:|:---------:|:----------------:|
> | **Hardness** |    **Method**    |           | $\underline{\text{Total Time}}$ |   **Obj.**    | **Inst. Infsb%** |              | $\underline{\text{Total Time}}$ | **Obj.**  | **Inst. Infsb%** |
> |     Easy     |       LKH3       |           |               46s               |     7.57      |      0.17%       |              |              1.8m               |   11.17   |      2.54%       |
> |     Easy     |  POMO*+PIP+LKH3  |           |               48s               |     7.48      |      0.20%       |              |              1.8m               |   10.46   |      0.01%       |
> |     Easy     | POMO*+PIP-D+LKH3 |           |               48s               |     7.47      |      0.32%       |              |              1.8m               |   10.52   |      0.00%       |
> |              |                  |           |                                 |               |                  |              |                                 |           |                  |
> |    Medium    |       LKH3       |           |               46s               |     13.36     |      0.67%       |              |              1.8m               |   20.42   |      26.66%      |
> |    Medium    |  POMO*+PIP+LKH3  |           |               48s               |     13.37     |      1.12%       |              |              1.9m               |   19.38   |      0.20%       |
> |    Medium    | POMO*+PIP-D+LKH3 |           |               48s               |     13.41     |      0.77%       |              |              1.9m               |   19.50   |      0.05%       |
> |              |                  |           |                                 |               |                  |              |                                 |           |                  |
> |     Hard     |       LKH3       |           |               44s               |     23.87     |      99.09%      |              |              1.9m               |     /     |     100.00%      |
> |     Hard     |  POMO*+PIP+LKH3  |           |               44s               |     25.65     |      2.99%       |              |              1.8m               |   51.39   |      15.80%      |
> |     Hard     | POMO*+PIP-D+LKH3 |           |               44s               |     25.68     |      3.23%       |              |              1.8m               |   51.37   |      6.24%       |
>
> ####
> ####
>
> **Table R9**: Results on LKH3 with the similar $\underline{\text {instance inference time}}$ limit as GFACS*+PIP(-D) (+LKH3) on $n=500$ (first 10 instances).
>
>
> |                   |                |     $n=500$      |              |
> | :---------------: | :------------: | :--------------: | :----------: |
> |    **Method**     | **Inst. Time** | **Inst. Infsb%** | **Gap to #** |
> |  LKH3 (Default)   |      26m       |      0.00%       |      #       |
> |       LKH3        |      6.5m      |      0.00%       |    0.59%     |
> |      GFACS*       |      6.4m      |      57.81%      |    21.32%    |
> |   GFACS*+PIP-D    |      6.5m      |      0.00%       |    11.52%    |
> |       LKH3        |      13m       |      0.00%       |    0.25%     |
> | GFACS*+PIP-D+LKH3 |      13m       |      0.00%       |    1.10%     |

---

### Official Review · Reviewer_DEtT · 2024-07-09

**Soundness:** 3
**Presentation:** 2
**Contribution:** 3
**Rating:** 6
**Confidence:** 5

**Summary:**

The paper  propose a novel Proactive Infeasibility Prevention (PIP) framework to advance the capabilities of neural methods towards more complex VRPs, and further investigates the Lagrange multiplier method for soft objective in VRPs, presenting the PIP (& PIP-D) for the hard case where Lagrange multiplier method difficult to find feasible solution. And the experiments show that the PIP is prior to Lagrange multiplier method and has generality to TSPTW and TSPDL.

**Strengths:**

This paper contributes to the solution of the soft constraint objectives of VRPs.
The author's perspective on the problem is also good; that is, the constraint itself is an NP-hard problem, which was not mentioned in reference [8], and the description of why the constraint itself is NP-hard is very clear, that is, because of the irreversible impact of the first selected node on the selection of subsequent customer nodes.

**Weaknesses:**

The author's method actually addresses the shortcomings of the Lagrange multiplier method, but it is not mentioned in the Introduction, and the problems raised in the Introduction can actually be solved only using the Lagrange multiplier method.
The method involves complex calculations, particularly when integrating the Lagrangian multiplier and PI masking, which can be computationally intensive​.
The paper does not thoroughly address the scalability of the proposed method when applied to extremely large datasets, which may limit its practical application in some real-world scenarios​​.

**Questions:**

[1]What are the potential challenges and limitations in scaling the PIP method to larger and more complex combinatorial optimization problems?

[2]How can we further optimize or modify the proposed method to reduce computational complexity without sacrificing solution quality or feasibility?

**Limitations:**

yes

---

> ### Author Rebuttal · Authors · 2024-08-07
>
> We thank the reviewer for recognizing our work as being with good perspective and very clear description. We understand that the main concerns are the computational cost and applicability towards real-world scenarios. We hope our response below will address them.
>
> ---
> **[Our PIP addresses shortcomings of the Lagrange multiplier method (W1)]** For original neural solvers, constraint awareness merely comes from the feasibility masking. However, the masking itself is NP-Hard in more complex variants, necessitating the use of a Lagrange multiplier to make the models aware of the constraints, thereby guiding the policy optimization. However, its advantages diminish as problem complexity increases (see Figure 2 and lines 312-323, where we provide in-depth discussions for the pros and cons of the Lagrange multiplier). For instance, in Hard datasets, the infeasible rate of POMO* (i.e., only with the Lagrange multiplier) is 100%. However, when equipped with our PIP-D, it drops dramatically to 6.48%. Therefore, only using the Lagrange multiplier is not enough for solving complex VRPs, while our PIP can address this shortcoming. We will refine the introduction and clearly mention this point in the revised paper.
>
> ---
> **[Computational complexity (W2)]** We understand the reviewer’s concern, as the acquisition of accurate masking is NP-Hard. We have implemented several strategies to enhance computational efficiency of our PIP-D, including one-step approximation of the PI masking, auxiliary decoder (PIP-D) to avoid intensive calculations of PI masking during training, and sparse strategy for large-scale datasets. The empirical results verify their effectiveness even facing large-scale instances. Please refer to `General Response #1`.
>
> ---
> **[Scalability (W3)]** We acknowledge that scalability is a significant challenge in NCO, and numerous concurrent works [10, 18, 21, 23, 48-52] are exploring scalable algorithms. However, we would like to clarify that **practical applications of NCO require not only scalability but also the ability to handle complex constraints**.
> - **Complex Constraints**: Our PIP is an **early work** to address the complex constraint handling in VRPs, whose significance is acknowledged by the reviewer.
> - **Scalability and Generality**: Our PIP is **orthogonal to scalability research** and can be combined with scalable algorithms. We have demonstrated our framework's generality and scalability using GFACS [10] on TSPTW-500. Note that scalability limitations often arise from the backbone itself. Our method can solve large-scale instances as long as the backbone model is capable of doing so.
>
> ---
> **[Potential challenges and limitations in scaling PIP to larger and more complex COPs (Q1)]**
> - **Larger Scale**: The main challenge and limitation is the trade-off between computation and performance. As acquiring the accurate feasibility masking is costly on large-scale instances due to the NP-Hard nature, we have to make some approximations and reduction of the search space to decrease overheads, which may sacrifice the solution quality and feasibility. Although our PIP has implemented approximation methods (e.g. one-step PI masking and auxiliary decoder to replace its acquisition), and can be applied to scalable neural solvers (e.g. GFACS), we cannot guarantee consistent performance across all scales (as noted in lines 346-351). A more effective and efficient approximation technique may be beneficial. To further reduce overheads, we also discuss some possible ways in the response to the next question.
> - **More Complex COPs**: As noted in lines 346-351, our PIP framework may not universally improve *performance* across all VRP variants, necessitating further experiments on various COPs. Another potential limitation is the *adaptability* of the PIP framework to all complex VRPs. We identify two types of complex VRPs: those with NP-Hard masking (solvable by PIP) and those with non NP-Hard masking but with large optimality gaps due to complex constraints. For the latter, PI masking is no longer necessary since we can obtain the feasibility masking easily at each constructive step, but the Lagrangian multiplier and the auxiliary decoder still have effects. Please see Figure R1 of the attached PDF, where we explored the application of PIP to VRPBLTW.
>
> ---
> **[Further trials to reduce the computational complexity (Q2)]** Below strategies can be considered for further accelerating the training and inference of PIP:
> - **Apply sparse strategies to refine PIP calculations:**
>   - **Only consider top K neighbours.** We have implemented this strategy on GFACS. Results in Table 3 show that GFACS*+PIP-D maintains similar training and inference times as GFACS* on TSPTW-500 (i.e., 28.3h/6.5m vs. 28.1h/6.4m).
>   - **Employ a trainable heatmap to confine the candidate space of PIP calculations.** Recent heatmap-based methods have successfully reduced the search space during construction, and we see similar potential for heatmaps to confine the candidate space of PIP calculation.
> - **Couple with the state-of-the-art solvers (e.g. LKH3)**: Our PIP is empirically verified to be efficient due to its capability to obtain good and feasible solutions within a very short time (LKH3: 1.4d vs POMO*+PIP-D: 48s). By further coupling with LKH3, our PIP can even achieve the performance of LKH3 within a few hours (9h). Details refer to `General Response #2`.
> - **Fine-tune Lagrangian method (*) with PI masking**: We found that PI masking does not need to be calculated throughout the entire training process. Fine-tuning a pre-trained Lagrangian method (e.g., POMO*) with PIP for a few epochs will also deliver competitive results. Details can be found in Appendix D.3.
> - **Early stop of PI masking**: We empirically observed that infeasibility primarily occurs in the initial steps of the construction process. Hence, it is possible to employ PIP only during these early stages. Details can be found in Appendix D.4.

---

> > ### Author Response · Authors · 2024-08-12
> >
> > Dear Reviewer DEtT,
> >
> > Thank you once again for your insightful comments and helpful suggestions. As the deadline for author-reviewer discussions is approaching, we would greatly appreciate it if you could take a moment to review our rebuttal. Please let us know if you have any further questions or concerns. Thank you very much for your time.
> >
> > Best,
> > Authors

---

> > > ### Comment · Reviewer_DEtT · 2024-08-13
> > >
> > > Thank you very much for your thoughtful response and the additional analysis. Most of my concerns have been resolved, and my score has been raised.

---

> > > > ### Author Response · Authors · 2024-08-13
> > > > **Thank you for your support!**
> > > >
> > > > We greatly appreciate your time in reviewing our paper and reading the follow-up rebuttal! We're thrilled for your recognition of our work, and will include the additional analysis in the revised paper. Thanks a lot!

---

> ### Comment · Area_Chair_JCXx · 2024-08-12
> **Dear Reviewers DEtT and Seke:**
>
> The authors have provided extensive comments and new results in response to the criticisms raised in your reviews. Has this response addressed your main concerns? If not, are there additional questions you would like to pose? The author discussion period is scheduled to end tomorrow (Aug 15). Please respond.

---

### Official Review · Reviewer_azSu · 2024-07-12

**Soundness:** 3
**Presentation:** 2
**Contribution:** 3
**Rating:** 7
**Confidence:** 4

**Summary:**

This paper proposes a Proactive Infeasibility Prevention (PIP) framework to enhance the ability of neural methods to handle complex constraints in Vehicle Routing Problem (VRP).
The PIP framework integrates the Lagrangian multiplier to enhance constraint awareness and introduces preventative infeasibility masking to proactively guide the solution construction process. Additionally, an extended version called PIP-D employs an auxiliary decoder and two adaptive strategies to learn and predict masking information, reducing computational costs during training.
These methods were extensively tested on different levels of constraint hardness in the Traveling Salesman Problem with Time Window (TSPTW) and Traveling Salesman Problem with Draft Limit (TSPDL) variants. The results demonstrate that the proposed methods enhance the capabilities of neural methods, significantly reducing infeasibility rates and improving solution quality.

**Strengths:**

1.	Adaptability to Complex Constraints: The paper devises a method to apply machine learning to constraint-based problems, which have been challenging to handle with deep learning due to the difficulty in finding feasible solutions.
2.	Experimental Validation: Experimental results demonstrate that the proposed method can compute feasible solutions more effectively compared to traditional methods.
3.	Integration with Constructive Methods: The proposed approach can be combined with constructive methods, enhancing its practical applicability and flexibility.

**Weaknesses:**

1. Generality of the Trained Model: It is unclear whether the trained model generalizes well to unseen instances, raising concerns about its robustness and applicability to different problem settings.
2. Lack of Theoretical Justification for PIP: The paper does not provide a theoretical justification for the superiority of using the Proactive Infeasibility Prevention (PIP) method, leaving its theoretical advantages unproven.
3. Lack of Classical methods with time limit: There are no comparison with the classical methods (e.g., LKH3) with a time limit. I am concerned that LKH3 could find "good" feasible solutions within a short time (e.g., within 5 minutes).

**Questions:**

- Can the proposed idea be applied to methods other than constructive methods? If it can be used to guide the search to avoid infeasibility, it would be a more practical idea.
- Alternatively, after the (possibly infeasible) solution constructed by a constructive method, is it easy to recover a feasibility through neighborhood search? If so, can you compare this with the avoidance of infeasibility using PIP?
- Is the model constructed with PIP tailored to the training dataset? In other words, when using the model constructed with PIP for instances in different area, does the number of infeasible instances decrease compared to the model without PIP?
- Is it possible to use the PIP framework for more general VRP (e.g., with multiple vehicles or capacity constraints)?
- Are there any experimental comparisons between the proposed model and LKH3 with time limit?

**Limitations:**

The limitations of this work are discussed in the last section.

---

> ### Author Rebuttal · Authors · 2024-08-07
>
> We appreciate the reviewer for the positive and valuable comments. We are delighted that the reviewer found our approach adaptable to complex constraints, effective and flexible. We hope that the following response, along with additional experiments, will address remaining concerns.
>
> ---
> **[Generalization Evaluation (W1)]** We have evaluated the generalization of our PIP(-D) to the unseen real-world instances in datasets [76] with different scales, distributions (e.g. node distributions, time window distributions and width) and constraint hardness in the original Appendix D.6 (Table 7). The results consistently showcase that our PIP(-D) could significantly reduce the infeasible rate and improve solution quality on in-distribution (Table 1,2,3) and out-of-distribution (Table 7) datasets.
>
> ---
> **[Theoretical analysis of PIP (W2)]** We acknowledge that the NCO domain is mainly built on empirical superiority, prioritizing in closing gaps between neural and traditional solvers, but lacks theoretical support.
> - **Regarding empirical superiority**: Our PIP(-D) has been applied to various constructive methods, including both autoregressive (AM, POMO) and non-autoregressive (GFACS) models. We have conducted extensive experiments on TSPTW and TSPDL with different constraint hardness levels, from small problem scales (50, 100) for AM and POMO to large scales (500) for GFACS. These experiments demonstrate significant reductions in infeasibility rates and substantial improvements in solution quality.
> - **Regarding theoretical support**, we would like to note that each component of our PIP(-D) does have some theoretical support.
>   - **Lagrangian Multiplier Method**: we represent an early trial to incorporate the Lagrangian multiplier method into constructive VRP solvers by interleaving constraints into the reward function, transitioning from Eq. (2) to Eq.(3), both of which are theoretically proven to be equivalent in [8].
>   - **PI Masking and Auxiliary Decoder**: As illustrated in Figure 1, the feasibility masking (i.e., $n$-step PIP) in complex constraints is NP-Hard. While iterating over all future possibilities would make PI masking complete, it is computationally inefficient. Therefore, we approximate it with one-step PI masking, whose efficiency is validated in Table R3 and R4 of the attached PDF. To enhance training efficiency, we use an auxiliary decoder to further approximate one-step PI masking, avoiding the need to acquire it continuously during training (see lines 293-296).
>
> We agree with the reviewer that theoretical justification is significant and acknowledge the need for further theoretical development, which we leave as a future work.
>
> ---
> **[Comparison with LKH3 with time limit (W3, Q5)]** We follow the suggestion and add experiments on our PIP(-D) and LKH3 with various time limits (from 5s to 3m per instance). We exhibit a comprehensive comparison in Table R2 and Figures R2 and R3 of the attached PDF, where our PIP outperforms LKH3 within limited time budget and achieves slightly better solution quality while using only 27% of the inference time (9 hours vs. 1.4 days) by further coupling with LKH3. For details please refer to `General Response #2`.
>
> ---
> **[Can our PIP apply to methods other than constructive methods? (Q1)]** Yes, our PIP can potentially be applied to most NCO methods.
> - **Constructive Methods**: We have validated its effectiveness on both autoregressive (AM, POMO) and non-autoregressive (GFACS) methods.
> - **Iterative Methods**:
>   - Our PIP can **provide better initial solutions to enhance search efficiency**, as demonstrated in Table R2 of the attached PDF.
>   - While the logic of iterative search differs from construction, making one-step PIP masking less applicable, components of our PIP framework, such as **the Lagrangian multiplier method for constraint awareness and the auxiliary decoder for learning infeasibility**, can still guide the search to avoid infeasibility. We consider it as a promising direction for future work.
>
> ---
> **[Can the infeasible constructed solutions be recovered by local search? (Q2)]** We conduct additional experiments using LKH3 with different sources of initial solutions. The results indicate that local search methods can recover infeasible solutions. However, our PIP-D provides the most promising (good and mostly feasible) initial solutions for LKH3, enhancing its search efficiency and achieving SOTA performance (see **Table R2** in the attached PDF). This suggests the promising potential of our PIP-D to assist strong heuristic solvers in future work.
>
> ---
> **[Generalizability of PIP to different datasets in different domains (Q3, Q4)]** We would like to clarify that our PIP is not tailored to the training datasets since it is a generic framework with the potential to be applied to different backbone models; that is, what the backbones can do, our PIP adapted to them can also achieve.
> - **Regarding VRPs:** We have conducted extensive experiments on both in-distribution and OOD datasets of variants like TSPTW and TSPDL, covering various levels of constraint hardness (Easy, Medium, Hard). Our PIP consistently outperforms other baselines in these scenarios. Although PIP may not universally improve performance across all VRP variants, we followed the reviewer's suggestions and explored its application to VRPs with various complex constraints, such as VRPBLTW-50 (VRP with constraints of capacity, backhaul, duration limit, and time window). Results demonstrate that PIP significantly enhances solution quality for VRPs with complex constraints (Our gap 1.80% vs. POMO's gap 9.17%, see Figure R1 of the attached PDF for more details).
> - **Regarding a broader range of COPs:** Beyond VRPs, our PIP has potential applications in other domains, such as job shop scheduling, where operations need to be completed in a specific order. Infeasibility can be proactively prevented using PIP. We plan to explore these applications as future work.

---

> > ### Author Response · Authors · 2024-08-12
> >
> > Dear Reviewer azSu,
> >
> > Thank you once again for your insightful comments and helpful suggestions. As the deadline for author-reviewer discussions is approaching, we would greatly appreciate it if you could take a moment to review our rebuttal. Please let us know if you have any further questions or concerns. Thank you very much for your time.
> >
> > Best,
> > Authors

---

> > > ### Comment · Reviewer_azSu · 2024-08-12
> > > **Official Comment by Reviewer azSu**
> > >
> > > Thank you very much for your thoughtful response and the additional analysis. Most of my concerns have been resolved.
> > >
> > > I understand that the theoretical analysis of PIP and its application to iterative methods are considered future work.
> > > I agree that the proposed methods and its experiments presented in this paper are valuable, and I raised my score.

---

> > > > ### Author Response · Authors · 2024-08-12
> > > > **Thank you for your support!**
> > > >
> > > > We really appreciate your constructive comments and strong support for our paper. Thank you!

---

### Author Rebuttal · Authors · 2024-08-07

We sincerely appreciate your efforts and insightful comments. We are pleased that reviewers found our PIP(-D) framework to be **novel** (#DEtT, #Seke, #Xb4p), **with** **good perspective** (#DEtT), **practical applicability** (#azSu), **and** **effectiveness in addressing the critical constraint handling problem** (#azSu, #Xb4p). We also appreciate the positive feedback, where reviewers found our paper **logically clear** (#DEtT, #Xb4p, #Seke) and our experiments with various baselines **extensive** (#azSu, #Xb4p, #Seke). In this global response, we address the common concerns.

---
**[Computational cost and scalability]** Reviewers #DEtT, #Xb4p and #Seke raised questions regarding the computational cost and scalability. While we acknowledge that our framework introduces some unavoidable overhead to the backbone model, we believe it is acceptable for the following reasons:
- **The overhead is actually offset by its strong effectiveness.** Following the reviewer’s suggestion, we show that baselines (POMO and AM) struggle to solve the studied complex constrained VRP, even with prolonged runtimes similar to ours. To see this, we gather additional results in **Table R1** of the attached PDF and show that: 1) Incorporating Lagrangian multiplier (POMO* and AM*) may lead to some improvement (in Table 1 and 2 of main paper) but not for the cases in Table R1 under complex constraints and larger scales; 2) **Even with extended inference time** (by sampling more solutions and data augmentation)**, existing methods do not deliver any feasible solutions;** 3) In contrast, our PIP-D with 48s time, significantly reduces infeasibility from 100% to 6.48% compared to baselines running for 2.5m, and exhibits a optimality gap around only 0.3%. Furthermore, PIP can perform even better with more inference time.
- **We explored several efficient strategies to reduce computational costs in the new PDF, the main paper, and Appendix D.3 and D.4.** These efforts provide a more detailed discussion, and we will follow the comments to better clarify our discussion in the new paper. Below, we summarize them.
  - **1) One-step PI Masking**: Instead of simulating all future possibilities, we use one-step PI masking to approximate NP-hard feasibility mask and reduce computational cost (see new Table R3, R4 and our response `[PIP with different step numbers]` to Reviewer #Xb4p).
  - **2) Auxiliary Decoder (PIP-D)**: To reduce intensive PI masking calculations, we use an auxiliary decoder to predict one-step PI masking, enhancing training efficiency by 1.5x compared to PIP, especially as scale and constraint complexity increase (see Table 1 and lines 293-296 of the original paper).
  - **3) Sparse Strategy**: We incorporate sparse strategy (selecting top K neighbours) to handle large-scale problems more efficiently. For GFACS with $n=500$, our **GFACS\*+(PIP; PIP-D) demonstrates similar inference times to GFACS\* (6.5m vs. 6.4m)**, while significantly reducing infeasibility (57.81% vs. 1.56%; 0.00%) and improving solution quality (21.32% vs. 15.04%; 11.95%), showing great efficiency.
  - **4) Other strategies**: We also enriched the discussion for future work (kindly refer to our response `[Further trials to reduce the computational complexity]` to Reviewer #DEtT for details).
- **Lastly, we believe scalability for larger-scale and applicability to more complex real-world constrained VRPs represent both important directions in NCO.** While our primary focus is more on the latter in this paper, our PIP is an early work to enable backbone models to tackle complex constrained VRPs where masking is NP-hard. Moreover, our PIP is generic and can enhance a wide range of NCO models, as demonstrated with both autoregressive (AM, POMO) and non-autoregressive (GFACS) models in our experiments. As the scalability of the backbone model increases, our PIP framework will also improve.

---
**[Enhanced comparison between PIP and LKH3]** We thank the reviewers (#Xb4p and #azSu) for the insightful suggestions. Previously, we used LKH3 with default settings, and we now compare our PIP(-D) and LKH3 with various time limits (from 5s to 3m per instance) for comprehensiveness. Please refer to Table R2, Figures R2 and R3 in the attached PDF, **where our PIP indeed outperforms LKH3**. We will include these discussions in the revised paper. Below, we summarize the key results.
- **When given a limited budget (i.e., a few seconds, as mentioned by Reviewer #Xb4p), LKH3 performs significantly worse than our POMO\*+PIP-D on the studied complex variants.** As shown in Table R2, while LKH3 is powerful with state-of-the-art quality, its advantage diminishes with limited time. Our POMO*+PIP-D reduces the infeasibility rate from 53.11% to 6.28% with 0.9s time per instance, compared to LKH3 with 5.1s time.
- **PIP achieves state-of-the-art performance by further coupling with LKH3.** To leverage the strengths of both approaches, we used our PIP-D to provide better initial solutions for LKH3. This combination reduced the infeasibility rate from 53.11% to 0.21% and improved the objective from 51.65 to 51.25 within only a few seconds. Notably, initializing LKH3 with POMO*+PIP-D outperforms the default LKH3 setup (10,000 trials), achieving slightly better solution quality while using only 27% of the inference time (9 hours vs. 1.4 days).
- We also show the progress of objective value and infeasibility rate over inference times in Figures R2, R3 for clearer comparison.

---
**[Additional experiments]** We have conducted additional experiments in the attached PDF, which are summarized below.
* **Table R1:** Results of POMO models under various inference time.
* **Table R2:** Results of LKH3 and POMO*+PIP-D under various time limits.
* **Table R3, R4:** Results of PIP with different steps on two datasets.
* **Figure R1:** Results on VRPBLTW.
* **Figures R2, R3:** Progress curves of objectives and infeasibility rates over different inference time

---

### Decision · Program_Chairs · 2024-09-25

**Decision:**

Accept (poster)

**Comment:**

This paper introduces and analyzes a  Proactive Infeasibility Prevention (PIP) framework increasing the number of feasible solutions generated when solving complex VRPs, ultimately leading to better quality final solutions found. The paper has several strengths - the problem addressed is an important one, the PIP approach is novel, and a comprehensive experimental analysis (including the generation of several sets of new results during the rebuttal discussion period) demonstrate the advantage of the approach in dealing with VRP/TSP problems with complex constraints. Finally the paper is well-written and understandable.

There were two main perceived weaknesses of the paper. First, was the fact that the proposed method incurs significantly more computational expense  than previous approaches to addressing this problem. Given this fact, there were concerns that some of the experimental analyses that were performed did not adequately make a fair comparison against prior existing approaches.   These concerns led to the addition of several new sets of experiments during the author rebuttal period, which clarified and further highlighted PIP's ability to outperform existing approaches in solution quality, and also showed the ability to further boost solution quality when used in combination. A second observed weakness was the lack of any theoretical model justifying the PIP approach. Subsequent discussion here pointed out that component elements of the approach (e.g., use of Lagrangian relaxation) did have theoretical justification, and consensus was reached that development of a theoretical basis was an important area for future research. Please address all original reviewer concerns and  incorporate all of these new results in the final version of the paper.